



# Sensitivity analysis of attenuation in convective rainfall
# at X-Band frequency using the Mountain Reference Technique

Guy Delrieu, Anil Kumar Khanal, Frédéric Cazenave, and Brice Boudevillain

HMCIS Team, Institute for Geosciences and Environmental research (IGE),

UMR 5001 (Université Grenoble Alpes, CNRS, IRD, Grenoble-INP), Grenoble, France

**Abstract.** The RadAlp experiment aims at improving quantitative precipitation estimation (QPE) in the Alps thanks to X-band

polarimetric radars and *in-situ* measurements deployed in the region of Grenoble, France. In this article, we revisit the physics of propagation and attenuation of microwaves in rain. We first derive four attenuation – reflectivity (*AZ*) algorithms constrained or not by path-integrated attenuations (PIA) estimated from the decrease in return of selected mountain targets when it rains compared to their dry-weather levels (the so-called Mountain Reference Technique - MRT). We also consider one simple polarimetric algorithm based on the profile of the total differential phase shift between the radar and the mountain targets. The

central idea of the work is to implement these five algorithms all together in the framework of a generalized sensitivity analysis in order to establish useful parameterizations for QPE. The parameter structure and the inherent mathematical ambiguity of the system of equations make it necessary to organize the optimization procedure in a nested way. The core of the procedure consists in (i) exploring with classical sampling techniques the space of the parameters allowed to be variable from one target to the other and from one time step to the next, (ii) computing a cost function (CF) quantifying the proximity of the simulated

profiles and (iii) selecting parameters sets for which a given CF threshold is exceeded. This core is activated for series of values of parameters supposed to be fixed, e.g. the radar calibration error for a given event. The sensitivity analysis is performed for a set of three convective events using the 0°-elevation PPI measurements of the Météo-France weather radar located on top of the Moucherotte Mount (altitude of 1901 m asl). It allows estimation of critical parameters for radar QPE using radar data alone. In addition to the radar calibration error, this includes time series of radome attenuation and estimations

of the coefficients of the power-law models relating the specific attenuation and the reflectivity (*A-Z* relationship) on the one hand and the specific attenuation and the specific differential phase shift (*A-$K_{dp}$* relationship) on the other hand. It is noteworthy that the *A-Z* and *A-$K_{dp}$* relationships obtained are consistent with those derived from concomitant drop size distribution measurements at ground level, in particular with a slightly non-linear *A-$K_{dp}$* relationship ($A = 0.275\ K_{dp}^{1.1}$). X-Band radome attenuations as high as 15 dB were estimated, leading to the recommendation of avoiding the use of radomes for remote sensing

of precipitation at such frequency.

Correspondence to: guy.delrieu@univ-grenoble-alpes.fr



## 1. Introduction

Estimation of atmospheric precipitation is important in a high mountain region such as the Alps for the assessment and management of water and snow resources (drinking water, hydro-power production, agriculture and tourism) as well as for
prediction of natural hazards associated with intense precipitation and snowpack melting. In complement with *in-situ* raingauge networks and snowpack monitoring systems, remote sensing using ground-based weather radar systems has a high potential that needs to be exploited but also a number of limitations that need to be surpassed. A first dilemma is related to the choice of the altitude of the radar setup with a compromise to be found between maximizing the visibility of the radar system(s) at the regional scale and increasing the representativeness of the measurements made in altitude with respect to precipitation
reaching the ground, especially during cold periods. A second dilemma is the well-known detection / resolution *versus* attenuation compromise, which is acute for weather radar frequencies. S-Band and C-Band frequencies (around 3 and 5 GHz, respectively) are traditionally preferred in continental-wide weather radar networks (Serafin and Wilson, 2000; Saxion et al. 2011, Saltikoff et al. 2019) for their appropriate precipitation detection capability and their moderate sensitivity to attenuation. In Europe, MeteoSwiss has the longest-standing experience in operating such a C-Band weather radar network in high-
mountain regions (Joss and Lee, 1995; Germann et al. 2006; Sideris et al. 2014; Foresti et al. 2018). Implementation of radars operating at the X-Band frequency (~9-10 GHz) has also been proposed in the last decades for research and operational applications at local scales, e.g., for precipitation monitoring in urban areas and/or in mountainous regions (Delrieu et al. 1997; McLaughlin et al. 2009; Scipion et al. 2013; Lengfeld et al. 2014, to name just a few). The renewed interest for the X-Band frequency, known for long to be prone to attenuation (e.g., Hitschfeld and Bordan 1954), is based on the promises of
polarimetric techniques (e.g. Bringi and Chandrasekar 2001; Ryzhkov et al. 2005) for attenuation correction (Testud et al. 2000; Matrosov and Clark, 2002; Matrosov et al. 2005; Matrosov et al. 2009; Koffi et al. 2014, Ryzhkov et al. 2014). Météo-France has chosen to complement the coverage of its operational radar network ARAMIS (for Application Radar à la Météorologie Infra-Synoptique) in the Alps by means of X-Band polarimetric radars. A first set of three radars was installed in Southern Alps within the RHyTMME project (Risques Hydrométéorologiques en Territoires de Montagnes et
Méditerranéens) in the period 2008-2013 (Westrelin et al. 2012; Yu et al. 2018). An additional radar (MOUC radar, hereinafter) was installed in 2014 on top of the Mount Moucherotte (1901 m) that dominates the valley of Grenoble. The RadAlp experiment (Khanal et al. 2019; Delrieu et al. 2020) is a contribution to research aimed at improving quantitative precipitation estimation (QPE) based on the Météo France MOUC radar, complemented by a suite of sensors installed on the Grenoble valley floor at the Institute for Geosciences and Environmental research (IGE, 210 m asl). This includes the IGE research X-
Band polarimetric radar named XPORT, a K-Band Micro Rain Radar (MRR) and *in-situ* sensors (disdrometers, raingauges).

The present article aims to show that mountain returns can be useful for the parameterisation of QPE algorithms for weather radar systems operating at attenuating frequencies in mountainous regions. It is part of a series of contributions devoted to the Surface Reference Technique proposed for spaceborne radar configuration (Meneghini et al. 1983; Marzoug and Amayenc, 1994; and more recently Meneghini et al. 2020) and its transposition to ground-based radar configurations with the Mountain





Reference Technique (Delrieu et al. 1997, Serrar et al. 2000, Delrieu et al. 2020). Figures 1 and 2 illustrate our point. Figure 1 shows a map of dry-weather mountain returns of the MOUC radar. The configuration of the radars operated in the RadAlp experiment is recalled in the insert; note that only the MOUC radar data is used in the current study. The measurements are taken at an elevation angle of 0° which corresponds to the lowest PPI of the volume-scanning strategy of the MOUC radar. The reflectivity data are averaged over a period of four hours; one PPI is performed at the 0°-elevation angle every five minutes.

We have selected 22 mountain targets corresponding to compact groups of gates in successive radials (3-6 typically; the radial spacing is 0.5°) and ranges (5-10 gates; the gate extent is 240 m) presenting a majority of dry-weather reflectivity values greater than 45 dBZ. The paths between the radar and the targets are free of beam blockages and present as few noisy gates (due to side lobes) as possible. In addition to the reflectivity map, the top graphs of Fig.2 display the co-polar correlation ($\rho_{hv}$) and the total differential phase shift ($\Psi_{dp}$) maps at 14:00 UTC on 21 July 2017 before the convective event that occurred that

day between 15:30 and 18:00 UTC. The $\Psi_{dp}$ map is essentially noisy at that time and the red colour in the $\rho_{hv}$ map, corresponding to values close to 1, highlights some small rain cells, in particular one in the south of the radar domain close to Target 22 (Grand Veymont Mount). The middle row maps correspond to the occurrence of intense precipitation over the city of Grenoble at 16:05 UTC. A peak of 40 mm h$^{-1}$ in ten minutes was recorded at that time by the raingauge located on top of the IGE building. The $\Psi_{dp}$ map displays marked increasing radial profiles in the North-East (NE) direction. The $\rho_{hv}$ map

allows a good delimitation of the whole rain pattern and clearly shows the dominance of the mountain returns over the rain returns for most of the Belledonne and Taillefer targets. The most striking observation on the reflectivity map is the dramatic decrease of the mountain returns of Targets 1-10 in the NE sector which results with no doubt from the rain cell falling over the city of Grenoble at that time. This is a clear example of what will be termed as "along-path attenuation" hereinafter. On the bottom row of Fig. 2, which corresponds to the measurements made at 17:00 UTC, one can observe a similar strong along-

path attenuation in the NE direction in the $\Psi_{dp}$ map, associated with a second 40 mm h$^{-1}$ rainrate peak at the IGE site (see eventually the hyetograph in Delrieu et al. (2020), their Fig. 2). But more impressive is the general decrease of returns from all the mountain targets, associated with a rain cell occurring at the radar site. This is an example of so-called "on-site attenuation", related to the formation of a water film on the radome, combined with along-path attenuation in the immediate vicinity of the radar site.


The article is organised as follows. In the theoretical part (section 2), we find useful to revisit in some detail the physics of propagation and attenuation of microwaves in rain. We derive (section 2.2) four attenuation – reflectivity (AZ) algorithms constrained or not by path-integrated attenuations (PIA) estimated from the decrease in return of selected mountain targets when it rains, compared to their dry-weather levels. We also consider one simple polarimetric algorithm based on the profile

of the total differential phase shift between the radar and the mountain targets (section 2.3). The structure and interdependencies of the parameters are discussed in section 2.4. This leads to the description of the principles of the generalized sensitivity





analysis proposed for studying the physical model at hand (section 3.1). The results obtained are illustrated and discussed item by item in sub-sections 3.2.1-3.2.5. Concluding remarks and future work are presented in section 4.

## 2. Theory


### 2.1 Basic definitions and notations

Let us express the radar returned power profile $P(r)$ [mW] as:

$$P(r) = (C/r^2) \, Z(r) \, AF(r) \tag{2.1}$$

where $Z(r)$ [mm$^6$ m$^{-3}$] is the true reflectivity profile, $AF(r)$ [-] is the attenuation factor at range $r$ [km] and $C$ is the radar

constant. We suppose the measured reflectivity profile $Z_m(r)$ to depend both on the attenuation and on a possible radar calibration error denoted $dC$:

$$Z_m(r) = P(r) \, r^2/C = Z(r) \, AF(r) \, dC \tag{2.2}$$

In addition to the running range $r$, let us consider the range $r_0$ corresponding to the blind range of the radar system, eventually extended to the range where the reflectivity measurements start to be free of spurious detections due e.g. to side lobes.

The attenuation factor $AF(r)$ is expressed as the product of two terms:

$$AF(r) = AF(r_0) \, AF(r_0, r) \tag{2.3}$$

where $AF(r_0)$ is the on-site attenuation factor which, as discussed in the introduction, may result from two main sources: attenuation due to a water film on the radome and along-path attenuation due to precipitation falling between the radar site and range $r_0$.

As a classical formulation (e.g. Marzoug and Amayenc, 1994), we express the two-way attenuation factor as a function of the specific attenuation profile $A(r)$ [dB km$^{-1}$] through the following equation:

$$AF(r) = AF(r_0) \exp(-0.46 \int_{r_0}^{r} A(s) \, ds) \tag{2.4}$$

To go further, we have to introduce relationships between the radar measurables (specific attenuation and reflectivity) and the variable of interest for QPE, i.e. the rainrate $R$ [mm h$^{-1}$], which are assumed to be of power type with the following notations:

$$A = a_{AZ} \, Z^{b_{AZ}} \tag{2.5}$$

$$R = a_{RA} \, A^{b_{RA}} \tag{2.6}$$

$$R = a_{RZ} \, Z^{b_{RZ}} \tag{2.7}$$





The order used for the variables in equations 2.5-2.7 is meaningful since the specific attenuation profile is derived from the measured reflectivity profile, while the rainrate profile can be derived in a second step either from the specific attenuation profile or from the corrected reflectivity profile. Due to the well-known lower variability of the *R-A* relationship compared to the *R-Z* relationship, (2.6) should be preferred to (2.7) for the estimation of the rainrate profiles (Ryzhkov et al. 2014).

Let us now consider another particular range, denoted $r_m$, where estimates of the attenuation factor may be available. We use the following notation:

$$AF_m(r_m) = AF(r_m)\, dAF_m \tag{2.8}$$

where $AF(r_m)$ is the true attenuation factor at range $r_m$ and the term $dAF_m$ represents a multiplicative error term. As illustrated in the introduction, such direct estimates of the attenuation factor can be obtained in mountainous regions from the MRT.

We frequently use hereinafter the notion of path-integrated attenuation (PIA), in units of dB, defined as:

$$PIA(r) = -10\, log_{10}(AF(r)) \tag{2.9}$$

Note that since $AF(r)$ is comprised between 1 (no attenuation) and 0 (full attenuation), the PIA subsequently takes values in the range of 0 (no attenuation) up to $+\infty$ (full attenuation). The PIAs at ranges $r_0$ and $r_m$ are denoted $PIA_0$ and $PIA_m$, respectively, in the following.

## 2.2 Formulation of the attenuation-reflectivity algorithms

The following mathematical developments are inspired by the works on rain-profiling algorithms in satellite measurement configuration (e.g., Meneghini et al. 1983; Marzoug, Amayenc 1994). The attenuation-reflectivity algorithms (*A-Z* algorithms) proposed in this section rely on two basic equations. The first one is the analytical solution of (2.4) when the power-law model (2.5) is supposed to represent perfectly the *A-Z* relationship. By taking the derivative of $AF^{b_{AZ}}(r_0, r)$ with respect to range $r$, one obtains:

$$d(AF^{b_{AZ}}(r_0, r))/dr = AF^{b_{AZ}}(r_0, r)(-0.46\, a_{AZ}\, b_{AZ}\, Z(r)^{b_{AZ}}) \tag{2.10}$$

Substitution of the true reflectivity by the measured reflectivity through (2.2) and integration between $r_0$ and $r$ yields:

$AF^{b_{AZ}}(r_0, r) = 1 - 0.46\, a_{AZ}\, b_{AZ}\, SZ(r_0, r) \,/\, (AF(r_0)\, dC)^{b_{AZ}}$

with:

(2.11)

$$SZ(r_0, r) = \int_{r_0}^{r} Z_m(s)^{b_{AZ}} ds.$$


The second equation is obtained by integrating (2.10) up to range $r_m$ and by introducing the attenuation factor estimate available at this range, yielding:

$$(AF(r_m)\,/AF(r_0))^{b_{AZ}} + 0.46\, a_{AZ}\, b_{AZ}\, SZ(r_0, r_m)/(AF(r_0)\, dC)^{b_{AZ}} = 1 \qquad (2.12)$$


We develop in the next sub-section four formulations of attenuation corrections for a supposedly homogeneous precipitation type, i.e. we assume the $a_{AZ}$ and $b_{AZ}$ coefficients to be constant along the propagation path. Each formulation filters out one of the four parameters $a_{AZ}$, $dC$, $AF(r_0)$ and $AF(r_m)$. Note that due to the mathematical expression of the intervening equations there is no possibility to filter out the $b_{AZ}$ parameter, which will be assumed to be constant, close to a value of 0.8 (Ryzhkov

et al. 2014), and to present a low sensitivity in the system of equations.

### 2.2.1 *AZhb* algorithm (independent of $PIA_m$)

This formulation is based on (2.11) only. In other words, it does not make use of $PIA_m$. By combining (2.11), (2.2) and (2.3),

one obtains a corrected reflectivity profile through the following equation:

$$Z_{AZhb}(r) = Z_m(r) \,/\, [(AF(r_0)\, dC)^{b_{AZ}} - 0.46\, a_{AZ}\, b_{AZ}\, SZ(r_0, r)]^{1/b_{AZ}} \qquad (2.13)$$

The specific attenuation profile follows from the use of the *A-Z* power-law model (2.5):


$$A_{AZhb}(r) = a_{AZ}\, Z_m^{b_{AZ}}(r) \,/\, [(AF(r_0)\, dC)^{b_{AZ}} - 0.46\, a_{AZ}\, b_{AZ}\, SZ(r_0, r)] \qquad (2.14)$$

This formulation is equivalent to the solution proposed early by Hitschfeld and Bordan (1954), hence the proposed name *AZhb*. It can be termed as a "forward algorithm" since only the measured reflectivities between range $r_0$ and the running range $r$ are

used for the correction at range $r$. The minus sign between the two terms of the denominator indicates that the denominator is not prevented to tend towards 0 when the $SZ$ cumulative term increases. This solution is subsequently known to be unstable and highly sensitive to calibration error, to inadequate values of the *A-Z* relationship coefficients and to on-site attenuation.





### 2.2.2 *AZC* algorithm (independent of *dC*)


The attenuation constraint (2.11) is now used to express $dC$ as:

$$dC = [0.46\, a_{AZ}\, b_{AZ}\, SZ(r_0, r_m)\, /\, (AF(r_0)^{b_{AZ}} - AF_m(r_m)^{b_{AZ}})\,]^{1/b_{AZ}}$$
(2.15)

which is introduced in (2.11) to yield:

$$AF_{AZC}^{b_{AZ}}(r_0, r) = [AF(r_0)^{b_{AZ}} SZ(r, r_m) + AF(r_m)^{b_{AZ}} SZ(r_0, r)]\, /\, AF(r_0)^{b_{AZ}} SZ(r_0, r_m)$$
(2.16)

The corrected reflectivity profile is then derived from (2.2), (2.3), (2.15) and (2.16) to read as:


$$Z_{AZC}(r) = Z_m(r)\, [AF(r_0)^{b_{AZ}} - AF(r_m)^{b_{AZ}}]^{1/b_{AZ}}\, /\, \{0.46\, a_{AZ}\, b_{AZ}\, [AF(r_0)^{b_{AZ}} SZ(r, r_m) + AF(r_m)^{b_{AZ}} SZ(r_0, r)]\}^{1/b_{AZ}}$$
(2.17)

Note that in the previous derivations, the expression of $dC$ given by (2.15) is used two times, first in the expression of

$AF_{AZC}^{b_{AZ}}(r_0, r)$ from (2.11) and then in the substitution of $dC$ in (2.2).

The specific attenuation profile follows from the use of the *A-Z* relationship (2.5):

$$A_{AZC}(r) = Z_m(r)^{b_{AZ}}\, [AF(r_0)^{b_{AZ}} - AF(r_m)^{b_{AZ}}]\, /\, \{0.46\, b_{AZ}\, [AF(r_0)^{b_{AZ}} SZ(r, r_m) + AF(r_m)^{b_{AZ}} SZ(r_0, r)]\}$$
(2.18)

In addition to their independence with respect to $dC$, it is interesting to note that both the attenuation factor profile and the specific attenuation profile provided by the *AZC* algorithm do not depend on the $a_{AZ}$ parameter. This parameter is however present in the expression of the reflectivity profile.


### 2.2.3 *AZα* algorithm (independent of $a_{AZ}$)

The attenuation constraint (11) is now used to express $a_{AZ}$ as:

$$a_{AZ} = [dC^{b_{AZ}}\, (AF(r_0)^{b_{AZ}} - AF(r_m)^{b_{AZ}})]\, /\, [0.46\, b_{AZ}\, SZ(r_0, r_m)]$$
(2.19)





which can be introduced in (2.11) to yield:

$$AF_{AZ\alpha}^{b_{AZ}}(r_0, r) = [AF(r_0)^{b_{AZ}} SZ(r, r_m) + AF(r_m)^{b_{AZ}} SZ(r_0, r)] / AF(r_0)^{b_{AZ}} SZ(r_0, r_m) \tag{2.20}$$


Equation 2.20 is actually identical to the $AF_{AZC}^{b_{AZ}}(r_0, r)$ expression (2.16). From (2.20), (2.2) and (2.3), the resulting corrected reflectivity profile can be expressed as:

$$Z_{AZ\alpha}(r) = Z_m(r) SZ(r_0, r_m)^{1/b_{AZ}} / \{dC [AF(r_0)^{b_{AZ}} SZ(r, r_m) + AF(r_m)^{b_{AZ}} SZ(r_0, r)] \}^{1/b_{AZ}} \tag{2.21}$$


One can note that $Z_{AZ\alpha}(r)$ is different from $Z_{AZC}(r)$ and that it depends on $dC$.

Next, it can be verified by using (2.21), (2.5) and (2.19) (a second time, for the necessary substitution of $a_{AZ}$) that the $AZ\alpha$ specific attenuation profile is identical to the $AZC$ specific attenuation profile given by (2.18) with:


$$A_{AZ\alpha}(r) = Z_m(r)^{b_{AZ}} [AF(r_0)^{b_{AZ}} - AF(r_m)^{b_{AZ}}] / \{0.46\ b_{AZ} [AF(r_0)^{b_{AZ}} SZ(r, r_m) + AF(r_m)^{b_{AZ}} SZ(r_0, r)]\}$$

$$\tag{2.22}$$

We emphasize that both the attenuation factor and specific attenuation profiles provided by the $AZC$ and $AZ\alpha$ algorithms are identical. Moreover they do not depend on the $a_{AZ}$ and $dC$ parameters. This is *a priori* a very interesting property of these algorithms. However, the reflectivity profiles provided by the two algorithms are different and, in particular, the reflectivity profile of the $AZ\alpha$ algorithm depends on $dC$ while the reflectivity profile of the $AZC$ algorithm depends on $a_{AZ}$.


**2.2.4 $AZ0$ algorithm (independent of $PIA_0$)**


The attenuation constraint (2.11) can finally be used to express $AF(r_0)^{b_{AZ}}$ as:

$$AF(r_0)^{b_{AZ}} = [0.46\ a_{AZ}\ b_{AZ}\ SZ(r_0, r_m) + (AF_m(r_m)\ dC)^{b_{AZ}}] / dC^{b_{AZ}} \tag{2.23}$$

which can be introduced in (2.11) to yield:

$$AF_{AZ0}^{b_{AZ}}(r_0, r) = \{0.46\ a_{AZ}\ b_{AZ}\ SZ(r, r_m) + AF(r_m)^{b_{AZ}} dC^{b_{AZ}}\} / \{0.46\ a_{AZ}\ b_{AZ}\ SZ(r_0, r_m) + (AF_m(r_m)\ dC)^{b_{AZ}}\}$$

$$\tag{2.24}$$



The resulting corrected reflectivity profile is:

$$Z_{AZ0}(r) = Z_m(r) \,/\, \{\, 0.46 \, a_{AZ} \, b_{AZ} \, SZ(r,r_m) + (AF_m(r_m) \, dC)^{b_{AZ}} \}^{1/b_{AZ}} \tag{2.25}$$

And the specific attenuation profile:


$$A_{AZ0}(r) = a_{AZ} \, Z_m(r)^{b_{AZ}} \,/\, \{0.46 \, a_{AZ} \, b_{AZ} \, SZ(r,r_m) + (AF_m(r_m) \, dC)^{b_{AZ}} \} \tag{2.26}$$

The *AZ0* algorithm has the simplest mathematical expressions among the three algorithms using the PIA constraint. It looks like a "backward algorithm" since the reflectivity and the specific attenuation profiles estimated at the running range $r$ depend

only on the measured reflectivities between ranges $r_m$ and $r$, while the *AZC* and *AZα* algorithms make use of the entire measured reflectivity profile between $r_0$ and $r_m$ for the estimations at range $r$.

The + signs in the denominators of eq. 2.18, 2.19, 2.21, 2.22, 2.25 and 2.26 are indicators of the inherent stability of the three algorithms using the PIA constraint, unlike the *AZhb* algorithm.


**2.3 Formulation of a simple polarimetric algorithm**

In the present study, we are making a basic use of polarimetry with the derivation of a PIA profile, denoted $PIA_{\Phi dp}(r)$, from the profile of the total differential phase shift on propagation, denoted $\Phi_{dp}(r_0,r)$ [°]:


$$\Phi_{dp}(r_0,r) = 2 \int_{r_0}^{r} K_{dp}(s) \, ds \tag{2.27}$$

where $K_{dp}$ is the specific differential phase shift on propagation [° km$^{-1}$]. Assuming a power-law relationship between the specific attenuation and the specific differential phase shift on propagation, with:


$$A = a_{AK} \, K_{dp}^{b_{AK}} \tag{2.28}$$

and using Eqs 2.4 and 2.9 yields:

$$PIA_{\Phi dp}(r) = PIA_0 + 2 \, a_{AK} \int_{r_0}^{r} K_{dp}^{b_{AK}}(s) \, ds \tag{2.29}$$



This polarimetry-derived *PIA* profile can be related to the *PIA* profiles obtained by integrating the *AZ* specific attenuation profiles given by Eqs 2.14, 2.18 and 2.26 (equivalently, the $PIA_{\Phi dp}(r)$ profile could be derived as a function of range and related to the *AZ* specific attenuation profiles).


**2.4 Analysis of the parameters of the considered physical model**

Equations 2.11, 2.12 and 2.29 form a system of equations with seven parameters (or unknowns), namely the coefficients of the *A-Z* relationship ($a_{AZ}$, $b_{AZ}$), the coefficients of the $A - K_{dp}$ relationship ($a_{AK}$, $b_{AK}$), the radar calibration error ($dC$), the

on-site attenuation ($PIA_0$) and the path-integrated attenuation at range $r_m$ ($PIA_m$). Estimation of the rainrate profiles will require two additional parameters, e.g. the two parameters ($a_{RA}$, $b_{RA}$) of the *R-A* relationship. The prefactors and exponents of the so-called $Z - A - K_{dp} - R$ relationships are mutually dependent since they are determined by the shape, density and size distributions of the hydrometeors and their electromagnetic properties, largely driven by their solid *versus* liquid composition. These coefficients may vary considerably from one precipitation type to another. In addition, even for a given precipitation

type, the actual $Z - A - K_{dp} - R$ relationships present an inherent variability with respect to the power-law models, associated with the greater or lesser proximity of the particle size distribution (*PSD*) moments associated to each particular variable (e.g. the 6th order *PSD* moment for the reflectivity, the 3.67th order *PSD* moment for the rainrate). As an ultimate complexity, when for a given propagation path various types of hydrometeors are successively encountered (e.g. rain, melting precipitation, snow), it would be desirable to apply the appropriate coefficients for the different precipitation types… provided one is able

to determine them. As a simplification in the present work, we will be considering a homogeneous precipitation type (convective rainfall). Because of the mathematical form of the equations at hand and the likely mutual dependence of the exponents and prefactors of the power-law models, we will assume the exponents of the *A-Z* and the $A - K_{dp}$ relationships to be constant for all the considered events while the prefactors will be allowed to vary for each single target and time step. The question of the *R-A* conversion is left aside in this study.


From a physical point of view, the parameters $dC$, $PIA_0$ and $PIA_m$ are mutually independent and *a priori* independent of the coefficients of the $Z - A - K_{dp} - R$ power-law models. It seems reasonable, and this is done in the following simulations, to assume the radar calibration error to be constant for a given precipitation event. Regarding on-site attenuation, Frasier et al. (2013) made a synthesis of previous theoretical and empirical studies, and provided an empirical model based on the

comparison of the measurements of two X-Band radar systems in the French Southern Alps, one equipped with a radome and the other one being radomeless. From this article, we have devised two sampling strategies for the parameter $PIA_0$. The first sampling strategy is a simple random draw of $PIA_0$ between 0 and 10 dB whatever the precipitation conditions at the radar site. The second one takes into account a dependence of $PIA_0$ on the measured reflectivity in the vicinity of the radar site, denoted $Z_0$. Based on Figure 5 in Frasier et al. (2013), we have fitted a coarse power-law model for X-band radome attenuation





on their experimental data, yielding $PIA_0^* = 0.0126\, Z_0^{1.6}$ with $PIA_0^*$ in dB and $Z_0$ in dBZ. Based on their Fig. 6 which shows important variations between the theoretical and empirical results proposed in the literature, we have defined a large range of lower and upper limits for the $PIA_0$ draws conditioned on $Z_0$ via $PIA_0^*$ (see Table 1). For the two sampling strategies, $PIA_0$ will be allowed to vary from one target to the next, i.e. in different directions, and from one time step to the next. The accuracy of the MRT-derived $PIA_m$ was studied by Delrieu et al. (1999) by comparing MRT estimates with direct measurements

obtained with a receiving antenna set up in the mountain range . They showed that (i) selecting strong mountain returns (typically greater than 45-50 dBZ) allows to mitigate the impact of precipitation falling over the target (negative bias), (ii) that a refined estimation of the so-called dry-weather baseline is required to account for the possible modification of backscattering properties of the mountain surfaces before and after the event and (iii) that the time variability of the dry-weather returns defines the minimum detectable PIA. These elements were accounted for in the present study by selecting strong mountain

targets, studying their dry-weather time variability (see also Delrieu et al. 2020) and subsequently defining the range of variation of the $dAF_m$ multiplicative error (Table 1).

## 3. Sensitivity analysis

### 3.1. Principle

The parameter structure analyzed in sub-section 2.4 led us to organize the optimization procedure in a nested way:

For a series of convective events, we assume the exponents of the $A$-$Z$ and $A - K_{dp}$ relationships to be constant;

For each event, we assume the radar calibration error to be constant. A simulation is performed for each combination of the $b_{AZ}$, $b_{AK}$ and $dC$ values listed in Table 1;

The simulation core is implemented as follows for each mountain target and each time step:

- The $Z_m(r)$ and $\Phi_{dp}(r)$ profiles between the radar and the mountain target are pre-processed. For each of the successive radials composing the target, this includes determination of gates affected by clutter in the region of the

mountain target and along the propagation path. This is done by considering both dry-weather mean values exceeding various thresholds (25 dBZ for significant clutter, 45 dBZ for a gate belonging to the mountain target) and by using the profile of the copolar correlation coefficient ($\rho_{hv}$) (Delrieu et al. 2020). The median $Z_m(r)$ and $\Phi_{dp}(r)$ profiles over the series of radials are then computed. The MRT $PIA_m$ is evaluated as the difference of the $Z_m$ mean values between the dry-weather baseline and the current time step, the mean being taken over all the gates composing the

target. The $r_0$ value is estimated as the range of the first gate for which four successive values (corresponding to a range extent of 960 m) exceeds a $\rho_{hv}$ value of 0.95. This last value is set as a threshold between precipitation and clutter / no precipitation (from the statistics presented in Khanal et al. 2019). The $Z_0$ value is computed as the product



of $1/dC$ (correction for the radar calibration error) and the mean reflectivity of the selected four successive gates if they are located within the first 2 km range; otherwise the $Z_0$ value is set to 0. The reader is referred to Khanal et al. (2022) for the most recent description of the fairly sophisticated procedure used for the $\Phi_{dp}(r)$ regularization based on the raw total differential phase shift profiles for all the radials associated with a given target. Note that a target is selected at a given time step for the following steps of the simulation if $PIA_m > 1$ dB and if a good quality index of the $\Phi_{dp}(r)$ regularization is obtained (Khanal et al. 2022).

- The Latin Hypercubes Sampling technique is then used to generate $N$ parameter sets (with $N = 200$ in the following) filling uniformly the parameter space composed of four parameters: the prefactors $a_{AZ}$ and $a_{AK}$, the on-site attenuation factor $AF(r_0)$ and the multiplicative error $dAF_m$ on the MRT attenuation factor. The central values and intervals of variation of these four parameters are listed in Table 1. It is noteworthy that the random draws are made on the dB-transformed ranges of parameters so that there are as many values below and above the central value, e.g. as many values between 0.15 and 0.3 on the one hand and between 0.3 and 0.6 on the other hand for the $a_{AK}$ parameter.

- After discarding unphysical parameter sets (e.g. those leading to $PIA_0 > PIA_m$), the five algorithms are implemented for all the remaining sets. A cost function (CF) is evaluated in order to measure the convergence / proximity of the five simulated profiles for each parameter set. The following *CF* was found to be appropriate:

$$
\begin{aligned}
CF = Mean(&NSE(Z_{AZhb}(r), Z_{AZC}(r)), \\
&NSE(Z_{AZC}(r), Z_{AZ\alpha}(r)), \\
&NSE(Z_{AZC}(r), Z_{AZ0}(r)), \\
&NSE(Z_{AZ\alpha}(r), Z_{AZ0}(r)), \\
&NSE(PIA_{AZC}(r), PIA_{\Phi dp}(r)), \\
&NSE(PIA_{AZ0}(r), PIA_{\Phi dp}(r)))
\end{aligned}
$$

(3.1)

where *Mean* stands for "the mean value of" and *NSE* is the Nash-Sutcliffe efficiency (Nash and Sutcliffe, 1970) between the two profiles indicated between brackets. The *NSE* criterion, or efficiency, is quite popular in hydrological sciences. It is employed in the context of parameter optimization since it has the definite advantage of being sensitive to both the average values and the correlation of the compared data series. Note that $NSE = 1$ denotes perfect agreement between the two series. The first four terms of the *CF* allow measuring the convergence of the four AZ reflectivity profiles that are different from each other (unlike the specific attenuation profiles of the $AZC$ and $AZ\alpha$ algorithms, see section 2.2). Due to the inherent instability of the $AZhb$ algorithm, we consider the first *NSE* term in the computation of the *CF* only if $PIA_m < 10\ dB$. Indeed, this 10 dB value proved to be about the maximum value this algorithm is able to deal with, even with an almost perfect parameterization (Delrieu et al. 1999b). The last two terms of the *CF* are measuring the proximity of the polarimetric algorithm with the $AZC$ and $AZ0$ algorithms in terms



of the *PIA* profiles. Averaging *NSE* values computed for reflectivity and *PIA* profiles is acceptable since the ranges of variation of these two variables are of the same order of magnitude (note that this would not be the case for reflectivity and specific attenuation profiles). In the following, we have selected $CF_{th} = 0.8$ as the threshold to be exceeded to consider a given parameter set as "optimal".


The acronyms *OPS* for "optimal parameter set" and *NOPS* for "number of optimal parameter sets" will be used hereinafter. The *NOPS* can be computed for a given target and time step and summed up for all the targets and time steps of an event and a series of events to yield a measure of the overall quality of a given simulation involving fixed parameters ($b_{AZ}, b_{AK}, dC$) and randomly drawn parameters ($a_{AZ}, a_{AK}, AF(r_0), dAF_m$) for each single target / time step using the *LHS* technique. We recognise

that the choice of the cost function and the "satisfaction threshold" are essentially subjective. They rely on the experience gained during the implementing of the simulation framework. Two elements can be mentioned on this subject: (i) accounting for the *AZhb* algorithm for low to moderate *PIAs* less than 10 dB proved to be a good option owing to the strong sensitivity of this algorithm on the calibration error; (ii) adding the polarimetric algorithm and the subsequent last two *NSEs* in the *CF* allowed to dramatically reduce the mathematical ambiguity of the physical model at hand. This ambiguity is indeed quite large

for the *AZ* algorithms considered alone, in particular with regard to the $dC$, $a_{AZ}$ and $AF(r_0)$ parameters.

### 3.2. Results

#### 3.2.1 Illustration for a given target and time step


Figure 3 gives an example of result of the core procedure for target 13 (T13) on 21 July 2017 16:05 UTC. For this case with a MRT PIA of 25.9 dB at a range of about 20 km, we get $\Phi_{dp}(r_0, r_m) = 71.5°$ and $Z_0 = 9.5$ dBZ. The optimal set of fixed parameters for the considered event is $dC^* = 0.4$ dB, $b_{AZ} = 0.78$ and $b_{AK}^* = 1.1$ (see next sub-sections). Since for the best *OPS*, all the reflectivity profiles overlap perfectly, the results presented in Fig.3 correspond actually to a less optimal set so

that one can see some differences between the solutions of the different algorithms. The set of optimal "*LHS*-sampled" parameters for this specific target / time step is $PIA_0^* = 0.46$ dB, $a_{AZ}^* = 1.01 \ 10^{-4}$, $a_{AK}^* = 0.34$ and $dAF_m^* = 0.99$. The *CF* value is 0.925, while the *CF* value obtained with the best *OPS* is 0.981. Note that 55 parameter sets overpassed the *CF* threshold value of 0.8 for this example, i.e. $NOPS = 55$. For this good (though not the best) OPS, the reflectivity profiles (Fig. 3a) call for the following comments. We have here a clear example of the inherent instability of the *AZhb* algorithm, which "blows

up" at a range of about 7 km for this parameterization. One should remember that this algorithm is not accounted for in the *CF* computation for such high *PIAs*, as explained in sub-section 3.1. The three other *AZ* algorithms give rather similar results. As a general behaviour (and in particular whatever the value of the on-site attenuation), we note that the optimal parameterizations lead to the convergence of the *AZC* and *AZ0* algorithms near the radar and to the convergence of the *AZα* and *AZ0* algorithms





on the other end of the profile. Fig 3b gives the solutions obtained in terms of specific attenuation profiles. The *AZhb* profile

is not drawn in this figure. As shown in sub-section 2.2, the *AZα* and *AZC* solutions are identical (represented in red) and

slightly different at long range from the *AZ0* solution. The comparison of the corrected and uncorrected profiles clearly shows

in this example the dramatic impact of attenuation as regard to both the underestimation of the first precipitation cell and the

non-detection of the second one. Fig. 3c displays the raw and processed $\Phi_{dp}$ profiles. For such a strong attenuation case, one

can see that the raw profile has little noise and no significant "bumps" that could sign a differential phase shift on backscattering

($\delta_{hv}$) contamination (Trömel et al., 2013). Finally, Fig. 3d allows comparison of the PIA profiles derived from the *AZC-AZα*

algorithms (identical solutions), the *AZ0* algorithm and from the $\Phi_{dp}$ profile. Although there are some differences, the overall

consistency between the three profiles is good.

### 3.2.2 Time series of optimal parameter values


Figure 4 presents the time series of the input variables and optimal parameters obtained for the best simulation of the 21 July

2017 convective event. The second sampling strategy making use of $Z_0$ (see Table 1) is considered for $PIA_0$ in this example.

We will come back in sub-section 3.2.5 on the relationship between $PIA_0$ (Fig. 4c) and $Z_0$ (Fig. 4a). The time series of the

medians of $PIA_m$ and $\Phi_{dp}(r_m)$ give an indication on the evolution of the storm intensity which was greater between 15:30

and 17:00 UTC with medians of about 20 dB and 60°, respectively. The interquartile ranges of these two variables are quite

large, as a result of both the variation of the radar-target distances (from 15 up to 40 km) and the precipitation variability as a

function of the azimuth, illustrated in Figs 1 and 2. The time evolution of the storm intensity is more marked on the *NOPS*

time series (Fig. 4f) with multiplicative factors in the range of 5 to 10 between the period 16:00-17:00 and the period 17:00-

18:00 UTC. Although for a given target, there is an increasing trend of *NOPS* when *PIAm* increases (not shown for the sake

of conciseness), this is also related to the higher number of targets "reached" (i.e. targets with *PIAm* values greater than 1 dB)

between 16:00 and 17:00 UTC. We draw the attention of the reader to the low *NOPS* values and to the singular values obtained

for the optimal parameters (Figs 4cde) at time step 17:00 UTC compared with the rest of the time series.  This is related to the

strong on-site attenuation already evidenced on Fig. 2 (bottom graphs), which will be discussed in more detail in sub-section

3.2.5.


Some explanations are required at this stage regarding the choice made in the present simulation exercise for the values and

ranges of variation of the prefactors and exponents of the $A - Z$ and $A - K_{dp}$ relationships. Estimations were obtained from

the processing of the drop size distribution (DSD) data collected with a PARSIVEL 2 disdrometer located at the IGE site. The

dataset includes 337 rainy days during the period April 2017 – March 2020. The raw DSD measurements have a time resolution

of 1 min. They are binned into 32 diameter classes with increasing sizes from 0.125 mm up to 6 mm. Various filters (Hachani

et al. 2017) were applied to discard anomalous data and, in particular to detect non-liquid precipitation, thanks to the falling





speed spectra. The volumetric concentration spectra were then computed at a 5-min resolution. DSD spectra with 5-min rainrate less than 0.1 mm h$^{-1}$ were discarded from the analysis. A dataset of about 14600 DSD spectra was thus obtained corresponding to all types of precipitation occurring in liquid phase in the Grenoble valley. As for the scattering model, we used the

CANTMAT version 1.2 software programme that was developed at Colorado State University by C. Tang and V.N. Bringi. The CANTMAT software uses the T-Matrix formulation to compute radar observables such as horizontal reflectivity, vertical reflectivity, differential reflectivity, co-polar cross-correlation, specific attenuation, specific phase shift, etc, as a function of the DSD, the radar frequency, air temperature, oblateness models and canting models for the raindrops as well as the incidence angle of the electromagnetic waves. The results presented herein were computed for the X-band frequency, a temperature of

10°C, the Beard and Chung (1986) oblateness model, a standard deviation of the canting angle of 10° and an incidence angle of 0° (horizontal scanning, like for the MOUC radar data).

Figure 5 illustrates the fittings of the $A - Z$ relationship obtained from a classical logarithm of base 10 transformation of the two variables. One can note that the scatterplot is well conditioned for deriving a power-law model in the sense that it does not

present any particular curvature. The models provide good fits for the highest values, which correspond to convective rainfall. The determination coefficient is high and the three regressions performed give subsequently parameter sets close to each other. Our choice is to select the least-rectangle fit since for these calculations based on DSD data, the two variables can be considered in an equal footing. From this analysis, we have chosen (Table 1) $b_{AZ} = 0.78$ as a fixed value for this exponent and $a_{AZ} = 1.0 \ 10^{-4}$ as the central value for the *LHS* sampling of the prefactor. Although the scatter of the points around the power-law

model suggests a possible range of variation of [-5, 5 dB] for the DSD-derived values, we have limited this range to [-3, 3 dB] in our simulations on the basis of the much bigger resolution volume of the radar and the assumption that the prefactor is constant throughout the reflectivity profile.

Figure 6 gives the results obtained for the $A - K_{dp}$ relationship. It can be seen that the scatterplot of the logarithmic of base

10 transformed variables (Fig. 6a) presents a significant curvature. Due to the important weight given to low and medium values, the fitted power-law models are clearly unsatisfactory for the highest values, which are of interest in the present study. We have therefore tested two other fitting techniques based on the natural values of the two variables (Fig. 6b). A linear fit with a 0-forced intercept yields $A_h = 0.32 \ K_{dp}$ which is consistent with linear relationships proposed in the literature (Schneebeli et al. 2013). However, once again, we note that this linear fit is not good for the highest values. The fitting of a

non-linear power-law model (*NLPL*) proves to be more satisfactory with $A_h = 0.30 \ K_{dp}^{1.1}$. Since the exponents estimated with the log-transformed data are close to 0.9, we have decided to perform several simulations with fixed values of $b_{AK}$ in the range [0.9 – 1.2] (see Table 1). Regarding the prefactor $a_{AK}$, we have considered a central value of 0.3 and a range of variation of [-3, 3dB], that is minimum and maximum values of 0.15 and 0.6, respectively.





Additional sensitivity tests can be performed on such DSD-derived relationships, including for instance the influence of the air / hydrometeor temperature, the precipitation type (e.g. stratiform versus convective rainfall), the DSD integration time step, etc. Concerning the last factor, we compared the results obtained for the 2-min and 5-min time steps and we found no significant influence on the coefficients of the power-law models, while the R² values were significantly downgraded for the 2-min time step (not shown here for the sake of conciseness). As for the precipitation type, we carried out a rough classification of the 337

events into stratiform and convective types, by considering an event as convective if a rainrate threshold of 10 mm h$^{-1}$ was exceeded for at least one 5-min time step during the event. As one would except from the scatterplots in Figs 5 and 6, significant differences appeared between the stratiform and convective $A - K_{dp}$ relationships whereas the $A$-$Z$ relationships were almost identical. This is an argument for keeping the exponent $b_{AZ}$ constant in the simulation procedure. Regarding the sensitivity on temperature, one possible extension of the present work could be to consider the temperature time series available for each

event at the IGE site in the scattering calculations. This would most likely result in an increase of the variability of the $A$-$Z$ and $A - K_{dp}$ relationships. As a classical concern, one may however wonder how the average temperature in the radar resolution volume could be estimated (Rhyzhkov et al. 2014). We chose herein to rely on the ability of the simulation procedure to deviate from the central values of the parameters and their ranges of variation to be large enough.

The time series of the prefactors $a_{AK}$ (Fig. 4d) and $a_{AZ}$ (Fig. 4e) exhibit similar behaviour with (i) median values close to the central values for the most intense part of the event between 16:00 and 16:50 UTC as well as between 17:15 and 17:45, (ii) significant deviations for the most on-site attenuation prone time steps (lower medians between 15:30 and 16:00 UTC and higher median at 17:00 UTC) and (iii) more erratic behaviour from one step to the next after 17:45 UTC at the end of the storm. The first point in the previous list is reassuring in terms of the possibility of using DSD-derived power-law models, and

particularly the DSD-derived $A$-$R$ relationship, for radar QPE. The second point is difficult to explain from a physical point of view. Coupled with the observation that the interquartile ranges are quite large, especially those of the $a_{AZ}$ parameter, we believe that the mathematical ambiguity (Haddad et al., 1995) of the system of equations at hand remains important. It is noteworthy to mention that the mathematical ambiguity of the $AZ$ algorithms alone is much larger (e.g. with larger interquartile ranges for the $a_{AZ}$ parameter). Introducing the constraints related to the polarimetric algorithm allowed to reduce it

dramatically.

### 3.2.3 Estimating the radar calibration error

In order to increase the robustness of the results, the simulation procedure was performed for three convective events that

occurred successively during summer 2017. Table 2 presents some characteristic features of these events. For all of them the melting layer (ML) altitude, determined with the 25°-elevation XPORT radar data by using the procedure developed in Khanal et al. 2019, was situated well above the altitude of the Moucherotte Mount radar, hence, there is no ML contamination of the considered radar data. The first two events were rather intense and similar in terms of total rain amount and maximum rainrate





at the IGE site, as well as in terms of the $PIA_m$ statistics based on the 22 mountain targets. The third one was a bit less intense.

To our knowledge, there was no occurrence of hail reported in the area of interest for these three events.

Figure 7 shows the evolution of *NOPS* for the three events separately and all together as a function of the fixed values of $dC$ listed in Table 1. The optimal values of the other fixed parameters are considered in these results with $b_{AZ} = 0.78$ and $b_{AK}^* = 1.1$. We note that the various curves are rather flat near their optimum values, e.g. with a ratio between the maximum *NOPS*

value and the nearest value, 0.4 dB apart, of 1.02 when the data of the three events are grouped. The overall sensitivity of the $dC$ parameter is clear however in the considered [-2, 2 dB] range, e.g. with a ratio of the maximum to the minimum *NOPS* values of 2.04 for the all-events curve. Although the global results tend to indicate a very slight underestimation of the measured reflectivities, one can note that the optimal $dC$ values vary from one event to the next. The 21 July 2017 event is different from the other two and the results suggest on the contrary a slight overestimation of the reflectivities in that case. We

find it difficult to know whether such variations in the electronic calibration of the radar from one event to the next could be physically realistic. In any case, an in-depth analysis of the time series showed that on-site attenuation could not be held responsible for this result.

### 3.2.4 Linearity of the $A - K_{dp}$ relationship


Figure 8 shows the simulation results for the series of $b_{AK}$ values listed in Table 1. We note a slight superiority of the simulation with $b_{AK} = 1.1$ compared to the one with $b_{AK} = 1.0$ in terms of the maximum value of the *NOPS* computed over the three events all together. This observation is also valid for each of the 3 events separately (not shown). The simulation with $b_{AK} = 0.9$ is clearly below the other two. For $b_{AK} = 1.1$, the log-transformed distribution of $a_{AK}$ computed over the three events is

nearly symmetrical with an average value of 0.275 and an interquartile range of nearly [-1, 1 dB]. Hence, we obtain in this study quite a remarkable agreement between the radar and DSD-derived $A - K_{dp}$ relationships for convective precipitation, with $A = 0.275 \, K_{dp}^{1.1}$ and $A = 0.30 \, K_{dp}^{1.1}$, respectively.

### 3.2.5 Radome attenuation


Coming back to Figure 4, we remind that the second sampling strategy making use of $Z_0$ was considered for the random drawing of $PIA_0$ values in this simulation. With $n = 3.0$, the crude model proposed in Table 1 yields upper limits of the $PIA_0$ sampling range of 3.0, 5.8, 9.2 and 13.1 dB for $Z_0$ values of 20, 30, 40 and 50 dBZ, respectively. One has to remark that such close-range reflectivity measurements are actually affected by radome attenuation. This may explain why estimated $PIA_0$

values are of the same order of magnitude for time step 17:00 UTC than for time steps between 15:30 and 15:55 while $Z_0$ values are about 10 dBZ higher in this second period. Thus the relevance of the $Z_0$ variable for detection and quantification of



on-site attenuation may remain limited for a radar equipped with a radome. Nevertheless, Figure 7 shows the comparison of the two $PIA_0$ sampling strategies making use or not of $Z_0$ (blue and red continuous curves), by reference to the $NOPS$ variable computed for the three convective events. This figure clearly evidences a superiority of the strategy taking into account, even in a crude manner, the precipitation conditions at the radar site.

Figure 9 gives two examples of the core procedure implementation in the case of severe on-site attenuation that occurred on 21 July 2017 at 17:00 UTC (Fig. 2 bottom graphs). The constraint on the $PIA_0$ sampling model was relaxed by considering $n = 10$ in the model of Table 1, that is upper limits of $PIA_0$ sampling range of 15.2 and 29.1 dB for $Z_0$ values of 30 and 40 dBZ, respectively. The returns from Target 04 (T04) allow to quantify both on-site attenuation and along-path attenuation due to precipitation falling over the city of Grenoble (NE sector) at that time (left-hand side example). At this range of about 40 km, we get $PIA_m = 47.9\ dB$ and $\Phi_{dp}(r_0, r_m) = 129.9°$. The returns from Target 19 (T19) located in the South-East sector (right-hand side) seem to be essentially affected by the precipitation conditions at the radar site. At this range of about 27 km, we get $PIA_m = 11.9\ dB$ and $\Phi_{dp}(r_m) = 12.2°$. This yields $PIA_m/\Phi_{dp}(r_0, r_m)$ ratios of 0.37 and 0.97 dB degree[-1] for the two targets, respectively. These values are clearly (especially the second one) well above the range of expected values for the slope of a supposedly linear $A - K_{dp}$ relationship (Schneebeli et al. 2013), which in addition to the generalized decrease of the mountain returns, is an indication of a significant on-site attenuation effect. The $dC$-corrected $Z_0$ values computed in the directions of the two targets are significantly different with 38.9 and 28.6 dBZ, respectively. One can observe the very good convergence of all the $AZ$ algorithms in both cases. In particular for T19, all the $AZ$ reflectivity profiles, including the $AZhb$ one, are perfectly matched. The agreement is also very good between the $PIA$ profiles of the $AZ$ algorithms and the one of the polarimetric algorithm, except for a very slight stall of $\mathrm{PIA}_{\Phi dp}(r)$ at a range of about 30 km for T04, likely due to disturbances associated with side-lobe effects (visible on the $\rho_{hv}$ PPI on top of Fig. 9).

For the two $OPS$ considered in Fig. 9, one gets $PIA_0$ values 10.1 and 10.8 dB. By considering the $PIA_0$ statistical distribution calculated over the optimal parameter sets of all the targets for the considered time step, one obtains a symmetrical distribution with a slightly higher mean value of 12.6 dB and a rather large interquartile range of 4.5 dB. The mean value increases somehow (13.5 dB) and the interquartile range decreases to 3.2 dB if the $PIA_0$ distribution is computed for targets 9-22 only, i.e. for targets with reduced along-path attenuation. It is worth noting that such statistics are not improved (e.g., interquartile range reduced) if one considers a more stringent satisfaction criterion (e.g. $CF_{th} = 0.9$ instead of $CF_{th} = 0.8$).

**4. Discussion and future work**

In this paper, we have started to implement a global approach to study the interactions between X-band microwaves and hydrometeors in a mountainous context. Emphasis was placed on the attenuation problem, which is known to be severe for the



frequency under consideration and essentially uncorrectable unless estimates of total attenuation are available at a distance from the radar. The RadAlp experiment allows us to obtain direct PIA estimates from the Mountain Reference Technique in some specific directions and undirect estimates from the processing of the profiles of total differential phase shift available for each radial. Although the polarimetric technique is *a priori* much more convenient to apply and has interesting characteristics (independence on radar calibration, on-site attenuation and partial beam blockages), it suffers from several limitations,

including (i) the fact that the $\Psi_{dp}$ profile is noisy for light precipitation, (ii) possible contaminations by the differential phase shift on propagation $\delta_{hv}$ (ii) possible impact of non-uniform beam filling and (iii) the need to specify the relationship between the specific attenuation and the specific differential phase shift which depends on hydrometeor types, temperature, and so on. In a similar way to the satellite configuration (e.g. the possibility to use the Surface Reference Technique in addition to the dual-frequency measurements at Ka and Ku Bands for processing the radar data of the GPM core platform ; Meneghini et al.

2020), we have proposed to take advantage of all the MRT and polarimetric measurements available to perform a generalized sensitivity analysis of the physical model of interest. In the simple case of convective precipitation, we obtained interesting results regarding the radar calibration, the radome attenuation and the coefficients of the $A - Z$ and $A - K_{dp}$ relationships. We note that for the estimated optimal radar calibration error, the $A$-$Z$ and $A$-$K_{dp}$ relationships derived from radar data are consistent with those derived from concomitant drop size distribution measurements at ground level, in particular with a slightly non-

linear $A$-$K_{dp}$ relationship ($A = 0.275\ K_{dp}^{1.1}$). This is reassuring regarding the relevance of microphysical data and scattering models for the radar QPE parameterization. We have deliberately left aside the question of the specific attenuation - rainrate conversion in this article. An interesting validation exercise to be performed consists in using the DSD-derived $A - R$ relationship for the conversion of the estimated specific attenuation profiles; then these radar rainrate estimates will be compared with the raingauge measurements available. Another outcome of the study is the quantification of X-Band radome

attenuation. Values as high as 15 dB were estimated, leading to the recommendation of avoiding the use of radomes for remote sensing of precipitation at such frequency. As an alternative, it would be desirable to develop specific sensors to detect / quantify the presence of water on the radome wall. The study showed that the measured reflectivity at the radar site is not a good predictor for radome attenuation. As a next step, we plan to extend the procedure to stratiform events with MOUC radar measurements made at times within or above the melting layer. The multi-angle, multi-frequency, polarimetric measurements

of the valley-based radars will be critical in this respect for the characterization of the ML from below (Khanal et al. 2019, 2022) and the mitigation of the mathematical ambiguity of the physical model of interest.




**Code and data availability**

There will be no problem to make available at a later stage the codes / data developed / used in this study, preferably through demands of collaboration to the authors.

**Conflict of interest**

The authors declare that they have no conflict of interest.


**Author contribution**

GD is the main contributor for this article (concept, theoretical developments, calculations, article writing, corresponding author). AKK (PhD student) and BB (assistant professor) are scientists contributing actively to the RadAlp experiment. They performed the internal review of the article. FC is a research engineer who built and keeps improving the XPORT radar, a key

instrument deployed in the RadAlp experiment. BB is also the HMCIS team leader and as such does a lot of work for the team members.

**Acknowledgements**

We are grateful to P.N. Gatlin (NASA Marshall Space Flight Center, Huntsville, AL) for providing the CANTMAT version

1.2 software developed at Colorado State University by C. Tang and V.N. Bringi, who we also thank. The RadAlp experiment is co-funded by the Labex osug@2020 of the Observatoire des Sciences de l'Univers de Grenoble, the Service Central Hydrométéorologique et d'Appui à la Prévision des Inondations (SCHAPI) and Electricité de France / Division Technique Générale (EDF/DTG).

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





**Table 1:** Values and ranges of variation of the attenuation model parameters in the sensitivity analysis

| Parameters fixed for a given simulation | | | |
|---|---|---|---|
| Parameter | Value(s) | | |
| $b_{AZ}$ | 0.78 | | |
| $b_{AK}$ | 0.9, 1.0, 1.05, 1.10, 1.15, 1.20 | | |
| $dC$ | [-2, 2 dB] with a step of 0.4 dB | | |
| **Parameters taken into account in the Latin Hypercubes Sampling for a given simulation** | | | |
| Parameter | Central value | Range of multiplicative coefficient of the central value (in dB) | Lower and upper limit |
| $a_{AZ}$ | $1.0\ 10^{-4}$ | [-3, 3 dB] | $[0.5\ 10^{-4}, 2.0\ 10^{-4}]$ |
| $a_{AK}$ | 0.3 | [-3, 3 dB] | [0.15, 0.6] |
| $dAF_m$ | 1.0 | [-1, 1 dB] | [0.79, 1.26] |
| $AF(r_0)$: sampling #1 | 0.316 | [-5, 5 dB] | $AF(r_0)$: [1.0, 0.1] corresponding to $PIA_0$: [0, 10 dB] |
| $AF(r_0)$: sampling #2 | $PIA_0^* = 0.0126\ Z_0^{1.6}$ $PIA_0^*$ [dB]; $Z_0$ [dBZ] $AF^*(r_0) = 10^{-PIA_0^*/10}$ | | Lower limits: $PIA_0^L = 0;\ A(r_0)^L = 1$ Upper limits: $PIA_0^U = n\ PIA_0^*$ $A(r_0)^U = 10^{-PIA_0^U/10}$ with $n = 3$ in results of Figs 3-4; 7-8 and $n = 10$ in results of Figs. 9-10 |




**Table 2.** Some characteristics of the three convective events considered in this study. The melting layer (ML) detection was performed with the 25°-elevation angle measurements of the XPORT radar using the algorithm described in Khanal et al.

(2019). The total rain amount and the maximum rainrate are recorded at the raingauge available at the IGE site at the bottom of the Grenoble valley. The $PIA_m$ statistics are derived from the MRT by considering all the 22 mountain targets and the 0° elevation data of the Moucherotte Mount radar.

| Date | Beginning (UTC) | End (UTC) | Minimum altitude of the ML bottom (m asl) | Total rain amount (mm) | Maximum rainrate in 10 min (mm h⁻¹) | Maximum $PIA_m$ value (dB) | Number of profiles with $PIA_m$ greater than a given value |
|------|-----------------|-----------|-------------------------------------------|------------------------|--------------------------------------|-----------------------------|------------------------------------------------------------|
| 21 July 2017 | 15:30 | 19:00 | 3000 | 35.2 | 42.0 | 59.8 | 11 (> 40 dB) |
| 8 August 2017 | 8:30 | 14:00 | 3700 | 27.9 | 48.0 | 63.4 | 20 (> 40 dB) |
| 31 August 2017 | 7:00 | 11:30 | 3200 | 19.9 | 15.5 | 17.5 | 8 (> 15 dB) |






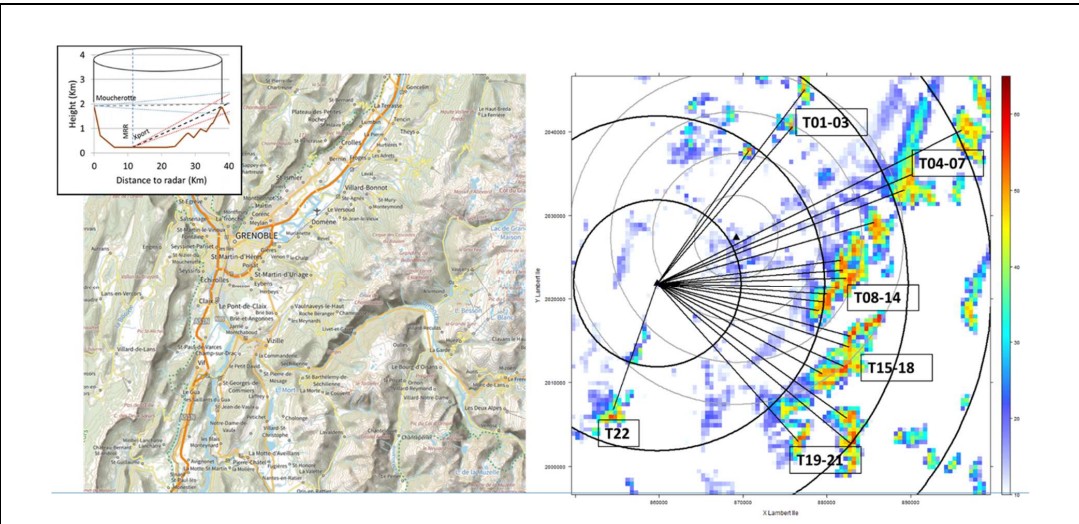

**Figure 1**: Left - 50x50 km² map of the region of Grenoble, France (from Geoportail, Institut Géographique National); Right – Reflectivity map of the X-Band weather radar located on top of the Moucherotte Mount (1901 m asl) in the Vercors massif. The radar is marked with a black triangle and circular range markers spaced by 10 km. The Cartesian map has a resolution of 500 m. The measurements were taken at an elevation angle of 0° during dry-weather conditions before the 21 July 2017 event. The radial lines indicate the azimuths and ranges of the 22 mountain targets used for the MRT implementation. Targets 1-3 are located in the Chartreuse Massif, targets 4-14 in the Belledonne Massif, targets15-21 in the Taillefer Massif and Target 22 in the Vercors Massif. In the background, the black triangle indicates the IGE site at the bottom of the valley (210 m asl). The grey circles with 5 km spacing indicate the coverage of the XPORT X-Band polarimetric radar whose measurements were used in the present study only for the detection of the melting layer.





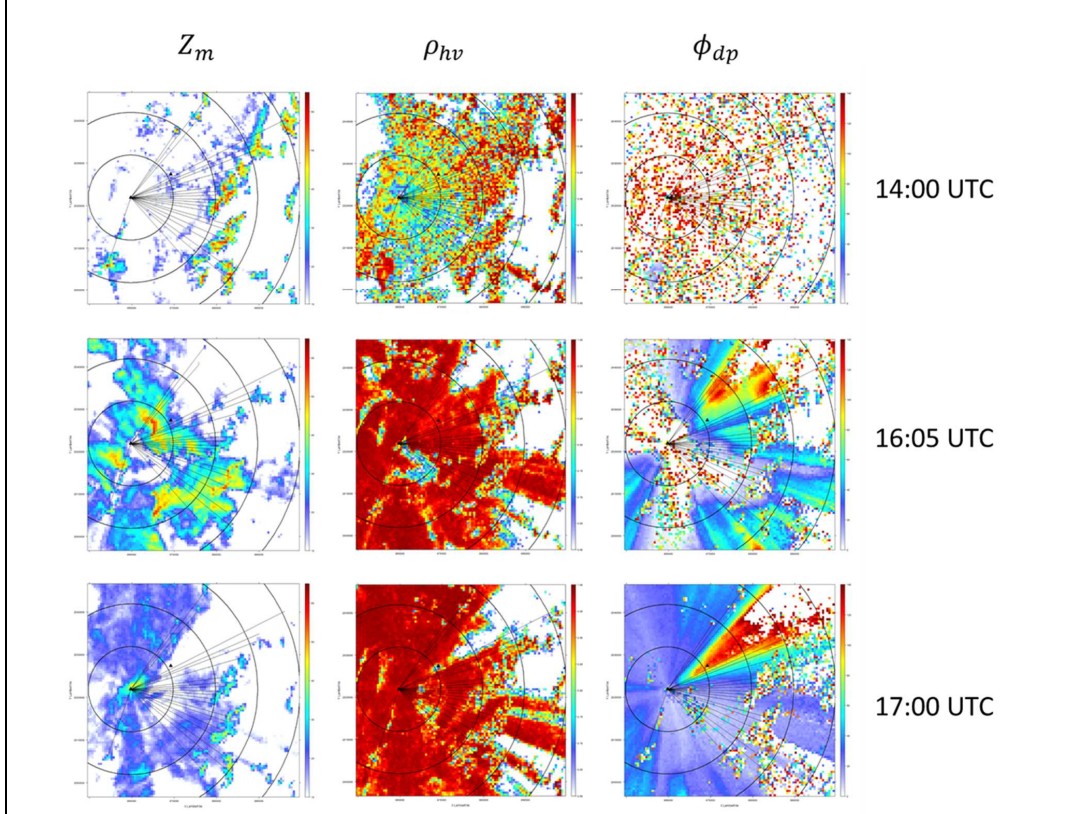

**Figure 2**: Examples of 0°-elevation PPIs of measured reflectivity (left), co-polar correlation coefficient (middle) and total differential phase shift (right) taken before (top) and at two moments of intense precipitation (middle and bottom) during the 21 July 2017 convective event. As in Fig. 1, the circular range markers of the Moucherotte Mount radar are spaced by 10 km.


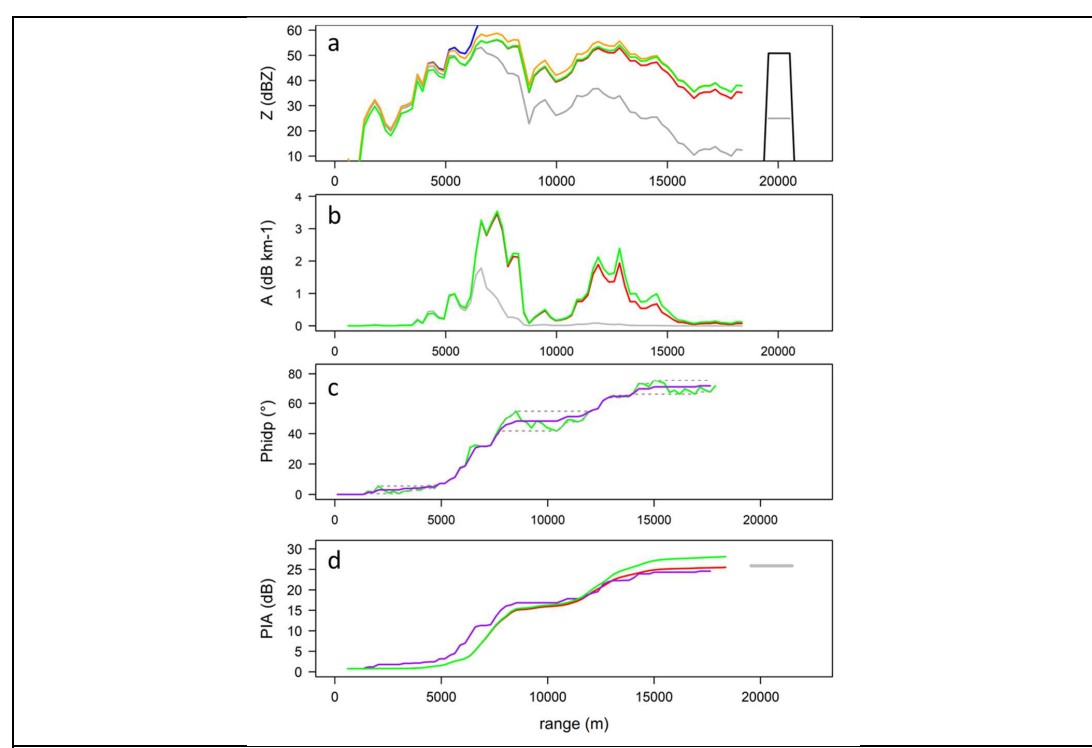

**Figure 3**: Implementation of the five algorithms (blue: *AZhb*; red: *AZC*; orange: *AZα*; green: *AZ0*; purple: $PIA_{\phi dp}$) for mountain target T13 during the 21 July 2017 convective event at 16:00 UTC using a near-optimal parameter set (see text for details). The results are displayed in terms of profiles of (**a**) reflectivity, (**b**) specific attenuation, (**c**) differential phase shift on propagation and (**d**) path-integrated attenuation. The grey profile in (a) is the measured reflectivity profile; the black and grey horizontal lines at range 20 km represent the mean dry-weather baseline and current reflectivities, respectively, of the mountain target. The resulting measured PIA value of 25.2 dB is reported in grey in (d). The grey profile in (b) is derived from the measured reflectivity profile by using eq. 2.5. The black line in (c) is the raw total differential phase shift profile and the grey dotted lines are the envelope curves used in the regularization procedure (Delrieu et al. 2020, Khanal et al. 2022).



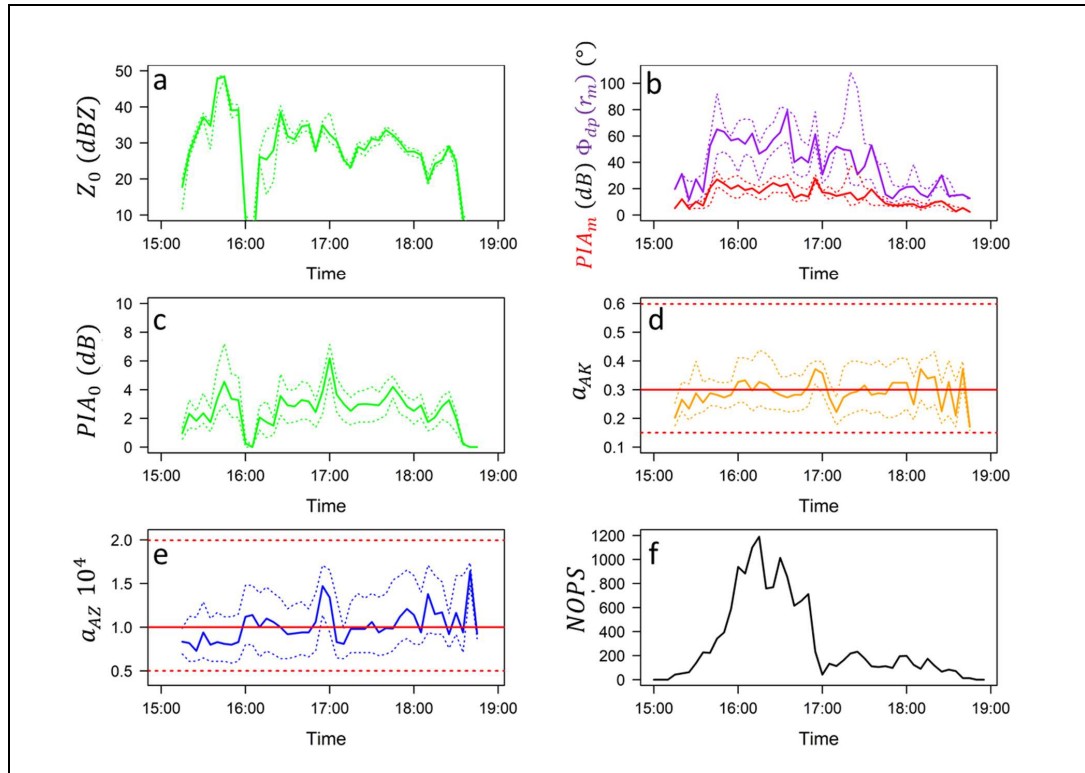

**Figure 4**: Time series of the input variables and optimal parameters for the best simulation obtained for the 21 July 2017 convective event. The optimal set of fixed parameters for this event is $dC^* = 0.4\ dB$, $b_{AZ} = 0.78$ and $b_{AK}^* = 1.1$. For each of the three considered input variables (**a**) $Z_0$; (**b**) $PIA_m$ (red) and $\Phi_{dp}(r_m)$ (purple), are displayed the median (continuous line) and the 25 and 75% quantiles (dotted lines) of their distributions over the 22 mountain targets. A similar representation is proposed for the *LHS* optimal parameters (**c**) $PIA_0$; (**d**) $a_{AK}$; (**e**) $a_{AZ}$, except that the distributions are established over all optimal parameters of all targets. The second sampling strategy making use of $Z_0$ (see Table 1) is considered for $PIA_0$ in this example. In (**d**) and (**e**), the dotted horizontal lines materialize the lower and upper limits consider in the *LHS* of the considered parameter. The time series of the number of optimal parameter sets cumulated over all the 22 targets (*NOPS*) is displayed in (**f**).


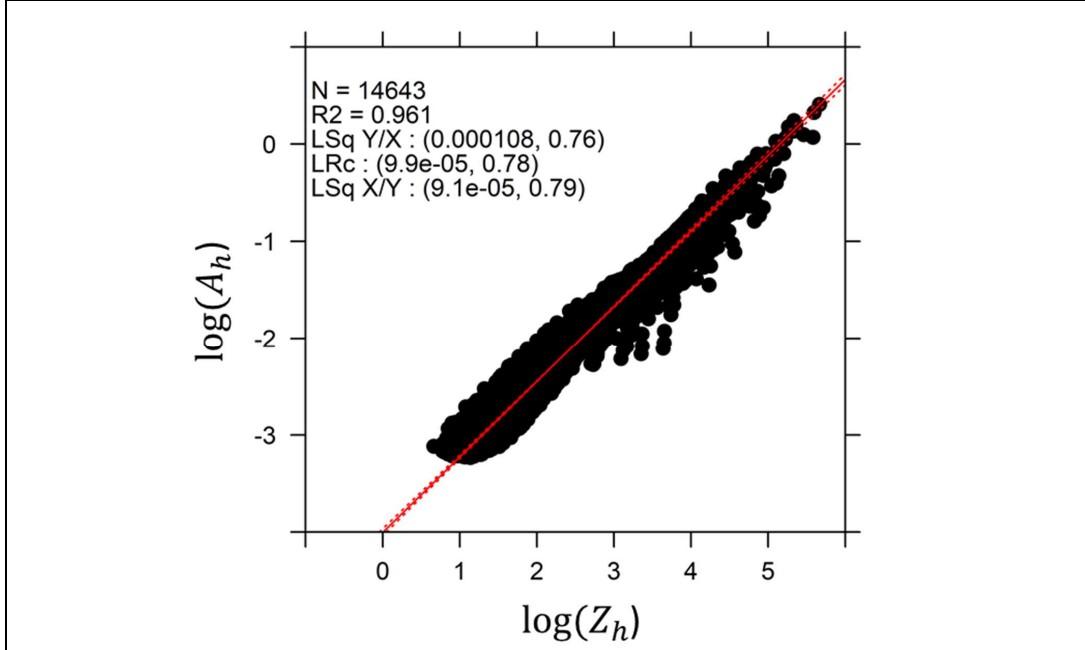

**Figure 5:** Results of the fitting of DSD-derived power-law models for the horizontal specific attenuation $A_h$ [dB km$^{-1}$] as a function of the horizontal reflectivity $Z_h$ [mm$^6$ m$^{-3}$] using a classical logarithmic of base 10 transformation of the two variables. Are given in the insert the number of points $N$, the square of the correlation coefficient ($R^2$) of the logarithmic regression, the prefactors and exponents of the resulting least-square regressions of the variable in ordinate versus the variable in abscissa (*Lsq Y/X*) and vice versa (*Lsq X/Y*) as well as the least-rectangle regression (*LRc*) which considers the two variables on an equal footing.


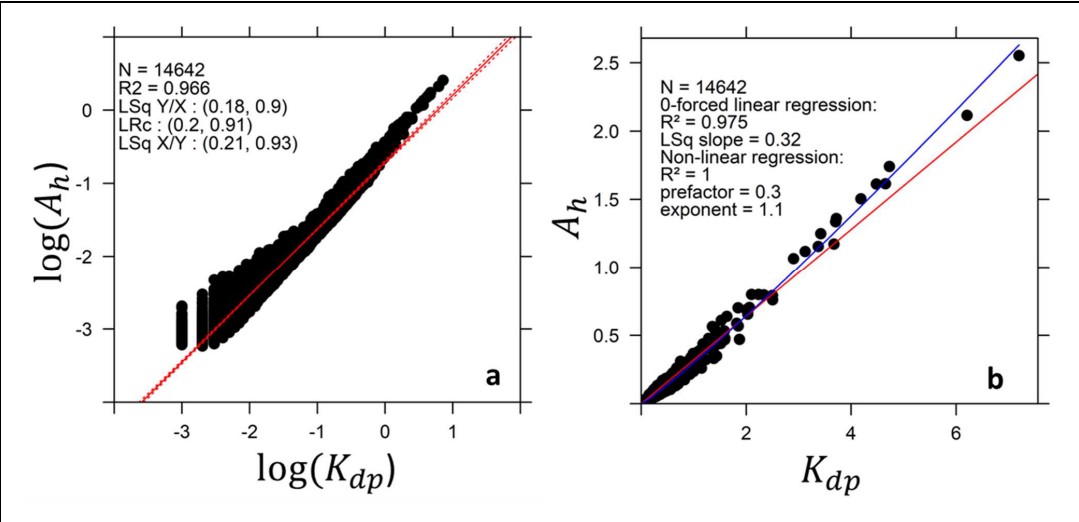

**Figure 6:** Fitting of DSD-derived power-law models for the horizontal specific attenuation $A_h$ [dB km$^{-1}$] as a function of the specific differential phase shift on propagation $K_{dp}$ [° km$^{-1}$] **(a)** using a classical logarithmic of base 10 transformation of the two variables (same comments as in Fig. 4 for this graph) and **(b)** using natural values of the two variables. The red line in **(b)** is the 0-forced linear regression with a slope equal to 0.32 and the blue curve is the non-linear fit of a power-law model with a prefactor of 0.30 and an exponent of 1.1.



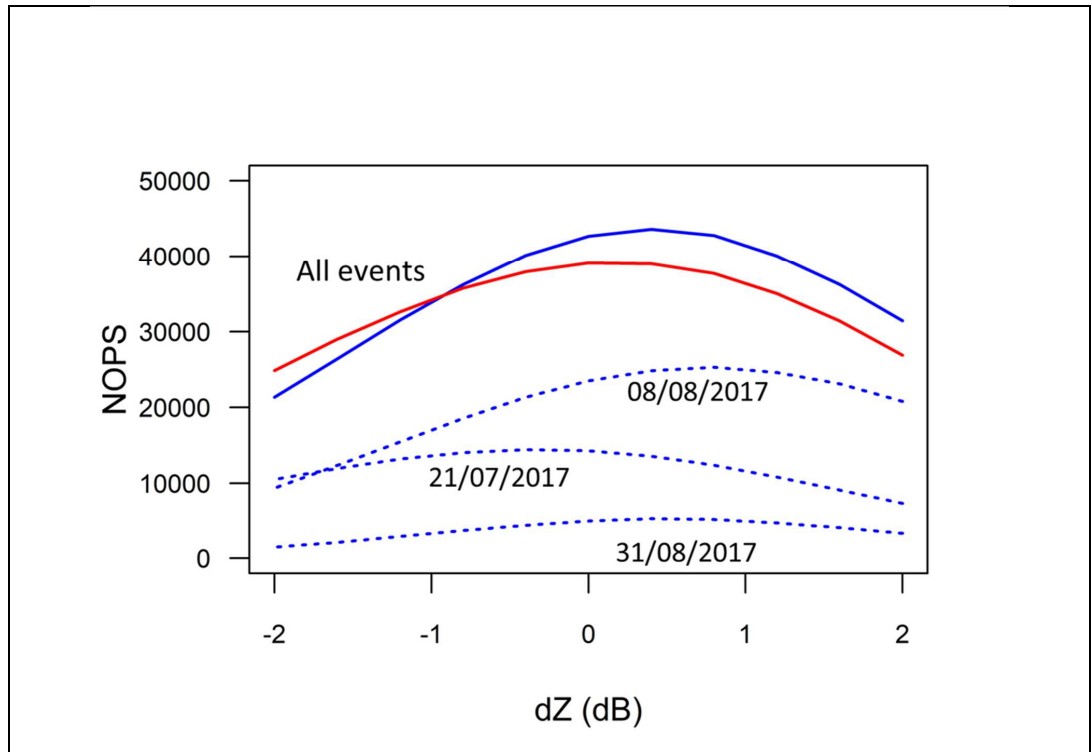

**Figure 7**: Evolution of the total number of optimal parameter sets (*NOPS*) as a function of the radar calibration error for three convective separately (dotted blue curves) and all together (solid blue curve) for the $PIA_0$ sampling strategy making use of $Z_0$ (sampling #2; Table 1). The solid red curve corresponds to the $PIA_0$ sampling strategy #1 (Table 1) for all events together. The variable $dZ$, equal to $-dC$, is used to feature the dBZ value to be *added* to the measured reflectivities for correcting for the calibration error. The other fixed parameters for these simulations are $b_{AZ} = 0.78$ and $b_{AK}^{*} = 1.1$.

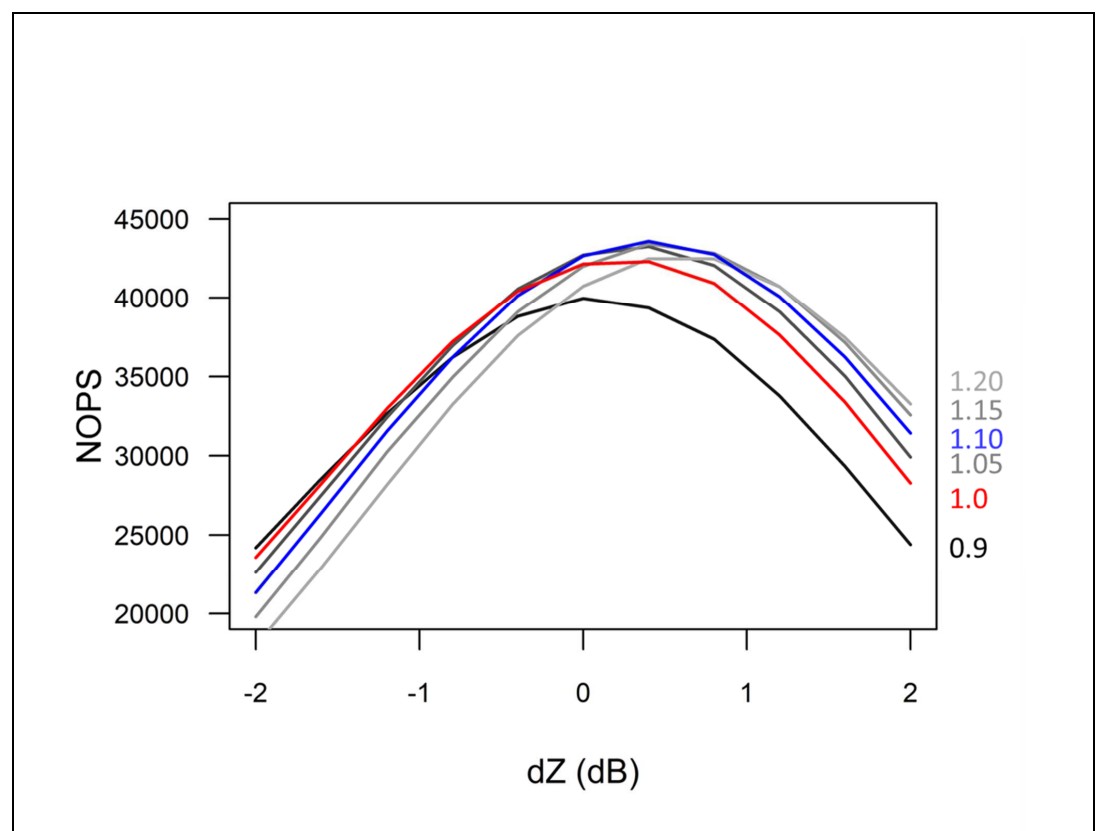

**Figure 8**: Evolution of the total number of optimal parameter sets (*NOPS*) computed for the three convective events all together as a function of *dZ* (see caption of Fig. 7) for various values of the exponent of the $A - K_{dp}$ relationship listed on the right-hand side of the figure. Like in Fig. 6, the red curve corresponds to $b_{AK} = 1.0$ and the blue curve to $b_{AK} = 1.1$.


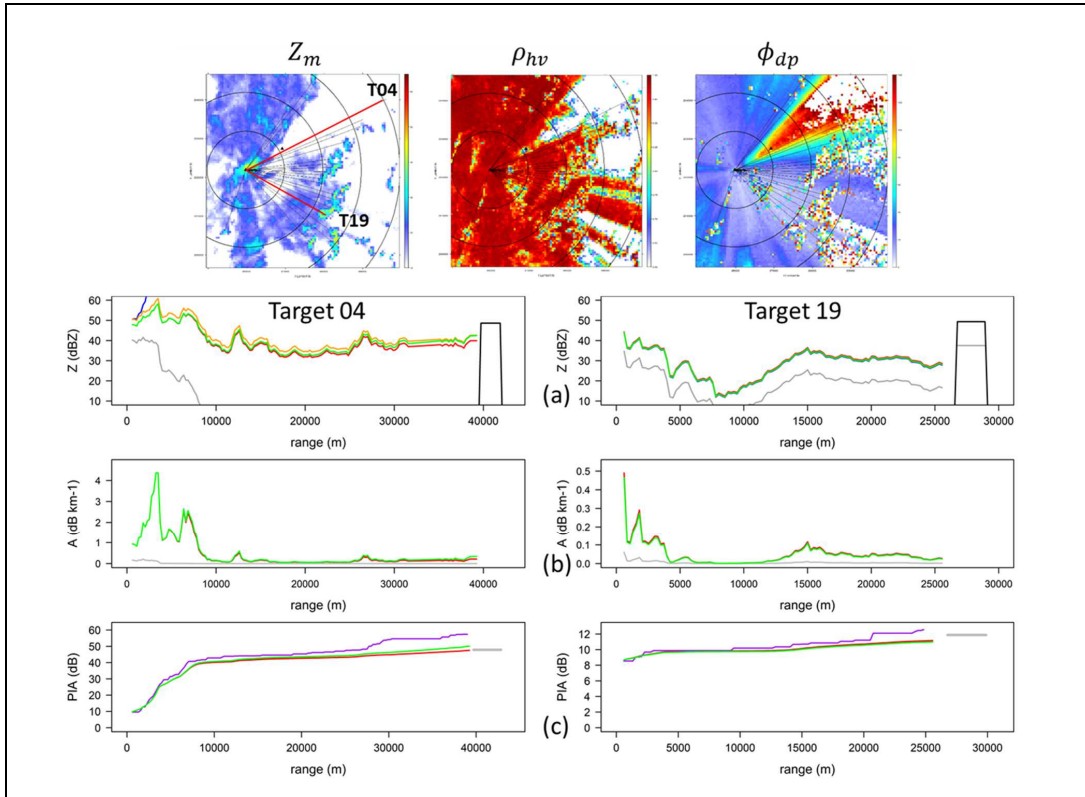

**Figure 9**: Implementation of the five algorithms with sets of optimal parameters (blue: *AZhb*; red: *AZC*; orange: *AZα*; green: *AZ0*; purple: $PIA_{\phi dp}$) on 21 July 2017 at 17:00 UTC for mountain target T04 with both along-path and on-site attenuation (left), as well as for target 19 with on-site attenuation mainly (right). The results are displayed in terms of profiles of (**a**) reflectivity, (**b**) specific attenuation, (**c**) path-integrated attenuation. In the upper images are displayed the PPIs of the measured reflectivity (with the indication of the position of the two targets in red), the co-polar correlation coefficient and the raw total differential phase shift.