# Peer review of "Sensitivity analysis of attenuation in convective rainfall at X-Band frequency using the Mountain Reference Technique"

_Atmospheric Measurement Techniques, 2022_

## Author Response (AR1)

In the end, following the suggestions of the reviewers and our own internal discussions, we have made a major revision of the article, which can be summarized as:

- We have moved part of the mathematical developments of the AZ algorithms of section 2.2 in an Appendix.
- We hope to have improved the motivation and description of the simulation framework (sub-sections 2.4 and 3.1).
- In particular, we have proposed and tested a new cost function (eq. 3.1), hopefully easier to understand, that proved to be as effective as the first one.
- All the simulations and related figures were subsequently re-done.
- The analysis of the DSD-derived Z-A-Kdp models was moved in section 2.4 (instead of 3.1); subsequently the figure order and numbering changed. Old Figs 5 and 6 become new Figs 3 and 4. References suggested by reviewer 3 were added.
- A new Figure (7) was added to complement the results of the 21 July 2017 case of Fig. 6 (old Fig. 4)
- We simplified the presentation of the on-site attenuation modelling by considering only the sampling strategy making use of Z0, and no more the "uniform" sampling of PIA0 between 0 and 10 dB whatever the precipitation occurring at the radar site.
- Figs 1 and 2 were improved as required by reviewer 3.
- The entire text was revised accordingly.

Our item by item replies to the reviewers, that have been posted early on the AMT site during the open-review process, are recalled hereinafter.

Again, we want to thank the three anonymous reviewers for their efforts in reviewing this article and the very valuable comments made.

The reviewer's comments/questions are recalled in blue, replies in black

This is a useful, well written paper that describes estimates of the attenuation-corrected radar reflectivity factor from a ground-based X-band radar using radar returns from the surrounding mountains as a path-attenuation constraint. I recommend publication.

Thank you for the time spent reviewing this article and the positive feedback.

One advantage of having a fixed radar and fixed targets, as opposed to airborne/spaceborne radar geometry, is that measurements of the reference target can be made before and after the rain event so that an assessment can be made as to how the target reflectivity might have changed during the event. Although dry and wet mountain targets can have different radar reflectivities, I would expect that a good assessment of the accuracy of the PIA estimate can be made.

The accuracy of the MRT-derived PIAs was studied by Delrieu et al. (1999) by comparing MRT estimates with direct measurements obtained with a receiving antenna set up in the mountain range. We showed at that time that (i) selecting strong mountain returns (typically greater than 45-50 dBZ) allows to mitigate the impact of precipitation falling over the target (negative bias), (ii) that a refined estimation of the so-called dry-weather baseline is required to account for the possible modification of backscattering properties of the mountain surfaces before and after the event and (iii) that the time variability of the dry-weather returns defines the minimum detectable PIA. These findings were accounted for in the present study with the selection of strong mountain targets (dry-weather reflectivities > 45 dBZ), a refined characterization of the dry-weather baselines and consideration of a 1 dB lower limit for "reliable" MRT PIAs.

A disadvantage of this geometry is that the mountain targets do not exist along all rays so that, I would imagine, some assumptions must be made to transfer information from estimates along rays/range-profiles with reference data to those without. Perhaps this is where the cost function becomes necessary.

The cost function is indeed one element of the proposed sensitivity analysis which uses the MRT PIAs available in some specific directions to optimize the parameters of the attenuation model (coefficients of the A-Z and A-Kdp relationships, calibration error, on-site attenuation). In a further step, this may allow implementation of attenuation correction algorithms in all directions, e.g. with polarimetric algorithms for significant/strong attenuations and/or with the AZhb algorithm for moderate attenuations (< 10 dB) for which the PHIdp signal is often too noisy.

Another difficulty is that the reflectivities of the targets are not all the same so the dynamic range of rain rates that are observable will vary from target to target. Similar issues arise with air/spaceborne platforms since the strength of the radar return from the surface depends on incidence angle and surface type. The authors note that most of the targets have a Zref value of at least 45 dB. For very strong target returns, I would guess that it's possible to see the mountain return even when the nearby rain signal is lost. (I realize that the authors address some of these issues in lines 311-331 and in some of their previous papers.)

The higher the dry-weather reflectivity of the mountain target, the wider the measurable PIA dynamic range. Maximum PIA values of about 60 dB were estimated in our study. For such big PIAs, both the reflectivity and the polarimetric signals are likely to be lost at some range between the radar and the target. So yes, it may be possible to quantify the PIA but not necessarily to reconstruct the entire rain profile. Note also that, due to the additivity of powers, considering mountain targets

with high dry-weather reflectivity (typically greater than the maximum expected precipitation reflectivity) is desirable to limit the effect of rain falling over the mountain. For example, a 50 dBZ rain falling over a 50 dBZ mountain target would result in a total signal of about 53 dBZ, introducing a 3 dB negative bias on the MRT PIA. While we would have preferred a 50 dBZ threshold in our study, consideration of a lower value (45 dBZ) was found necessary to get targets with rather homogeneous sizes. The dynamic range of the reflectivities of the selected mountain targets is [45 – 65 dBZ], therefore we expect a limited influence of rain falling over the targets, even for the convective cases considered in the article.

*What about rain that occurs beyond or above the ranges at which targets are present. Do the methods work well in these areas? Visual comparisons of the PPIs in Figs. 1 and 2 seem to indicate the existence of radar returns from rain beyond the mountain returns. These plots also seem to show some rays that contain multiple targets that are widely separated in range. Can these methods be generalized to rays having multiple reference targets?*

In the presented case study, we intentionally used the lowest elevation angle (0°) of the volume-scanning protocol of the Moucherotte radar to get the strongest mountain returns in order to obtain the most accurate PIA estimates. Applicability of the AZC, AZalpha and AZ0 algorithms is limited to profiles with a MRT PIA estimate at a given range rm (beginning of the mountain target). Hence for ranges above and beyond the mountain target, one must rely on measurements made at upper elevation angles and on AZhb and polarimetric algorithms, eventually constrained by the proposed parameter optimization method.

*The modified α (eq. 2.21) or C methods (eq. 2.17) depend on the unknown attenuation factor to range r0 whereas the final-value method (eq. 2.26) does not. This would appear to be an advantage of the final-value. However, it doesn't seem to be possible to apply the final-value method to rays that do not contain a reference target.*

Indeed the three formulations require a PIA estimate at range rm. Each of them "filters out" one of the three important sources of error associated respectively with the (rather subjective) choice of the α value, the determination of the radar calibration error and possible on-site attenuation. I find difficult to say whether one of the three formulations has an advantage over the others. The "philosophy" of our method is more to use these three formulations all together for estimating the α, dC and PIA0 values that lead to a convergence of their solutions (the cost function being a "simple" measure of this convergence).

*I had some difficulty understanding the motivation for the cost function given by eq. (3.1)…*

This is a critical point, explained in detail in section 3.1. Let's try to rephrase it in another way: we propose in fact a parameter optimization procedure based (initially) on four different mathematical formulations of the attenuation-reflectivity equations (4 AZ algorithms: AZhb, AZC, AZα and AZ0) accounting (or not for AZhb) for a MRT PIA estimate available at a given range rm:

1)      Using the "Latin Hypercubes Sampling" technique, we draw randomly sets of parameter values (α, dC and PIA0; but also an error term on the MRT PIA value) sampling uniformly the "parameter space".

2)      For each of these parameter sets, we compute the corrected reflectivity profiles given by the 4 formulations.

3)      We are happy when a given parameter set allows a satisfactory convergence of the solutions of the 4 algorithms. Measuring this convergence is the role of the cost function.

4)       Considering the resulting "optimal parameter sets" obtained for all the targets and all the time steps of an event and a series of events, we infer some values and trends on the calibration error, the coefficients of the AZ relationship and the radome attenuation, that can be used in a further step in the implementation of given algorithms.

But, as noted early (e.g. Haddad et al. 1995), the system of attenuation-reflectivity equations is prone to mathematical ambiguity, i.e. several combination of parameters (including non-physical values) may lead to the convergence of the solutions of the different algorithms. This is a fundamental limitation of our attempts to optimize the attenuation equation parameters (this is quite a frequent situation in environmental sciences…). We have found however that this mathematical ambiguity was significantly reduced when we took into account more information (more constraints) with a fifth algorithm based on polarimetric data (the Phidp profiles). This led to a complexification of the cost function (eq 3.1) but also to added results about the coefficients of the A-Kdp relationship.

Also, due to its "explosivity", we found necessary to limit consideration of the AZhb algorithm in the cost function to moderate PIAs (less than 10 dB).

…so let me ask the following question.  Assume that modified α's from, say N, mountain targets are obtained, at a given time step, and the mean is taken.  This modified mean α could then be used to obtain attenuation-corrected Z profiles over the full volume scan of the radar, including rays with no reference target.  Would these profiles be significantly different from the profiles obtained by minimizing the cost function?  The same procedure could be done for the C-adjustment approach but it would be difficult to interpret this physically since C should be independent of the viewing angle - unless this adjustable C could somehow account for radome loses that change with look-angle.

We had this kind of discussion in our section 2.4 about the analysis of the a priori values to be given to the parameters of the physical model at hand. In short:

1)       we assumed the radar calibration error to be constant for a given event;

2)       the PIA0 values were allowed to vary from one time step to the next and from one direction (target) to the next;  we took into account (or not) the Z0 value at the radar site as a proxy for significant radome attenuation;

3)       the MRT PIAm values were supposed to vary in a [-1, 1 dB] range around the measured value;

4)       acknowledging the dependency of the coefficients of the A-Z and A-Kdp relationships on the underlying drop size distribution, we choose to consider several fixed values for the exponents and to let the prefactors vary in a given range around central values. The a priori values of the exponents and central values of the prefactors were estimated from concomitant DSD measurements

5)       the optimal (a posteriori) parameter values were determined by considering the total number of optimal parameter sets for each simulation.

Again, this kind of approach probably wouldn't work for the final-value (Marzoug-Amayenc) method as the equation doesn't have an adjustable parameter.

not sure to understand. Both dC and alpha are parameters for the AZ0 algorithm…

In Fig. 3, results from 6 methods are shown but it's sometimes difficult to track the behavior of the individual methods.  For example, the HB estimate seems to diverge for ranges beyond about 6 km.

In fact, the blue line (HB) in panel a is only visible around 5 km; for closer ranges, it probably exists but is hidden by the other curves.

Yes, we acknowledge that the behaviour of the different algorithms is difficult to track since we are essentially looking for them to converge! This is effectively the case for the AZhb solution hidden by the others at range less than 5 km in Fig. 3. It is important to remind that for this profile with a 25 dB PIA, the AZhb algorithm was not accounted for in the cost function because of its inherent inability to deal with such great PIAs. The right panels of Fig. 9 present a case with full convergence of the 4 AZ algorithms for a profile with a PIA of about 10 dB, while the AZhb is also not considered in the calculations of the left panel (profile with a PIA of about 40 dB).

Z0 is defined at bottom of p. 10 as the measured reflectivity in the vicinity of the radar site, which is the range which is greater than the blind range and any clutter. If Z is the attenuation-corrected reflectivity at this range, then is the following equation correct: Z=Z0+PIA0? (where Z0, Z are in dBZ units).

Yes, in our calculations, Z0 value is just corrected for the supposed calibration error (dC) but not for the on-site attenuation. This is in part why this value is a poor predictor for PIA0. This could be improved by implementing some iterations in the (already heavy) calculations.

It seems that the phi-DP measurement has greater information content than the MRT in the sense that it provides an estimate of path attenuation to any range whereas the mountain return yields only a single path-attenuation estimate between the radar and the target. Is it correct to say that the phi-DP used in this paper is the value near the reference target? Couldn't it be used as a continuous variable to help validate the MRT estimates or is it too noisy? (Not sure if I'm making myself clear: if, at an arbitrary range, r, the phi-DP is used to estimate the two-way attenuation to that range, A(r), then Z(r)=Zm(r)+A(r), where Z, Zm are in dBZ units.)

Yes, the polarimetric measurements are much more convenient in the sense they allow PIA estimation for any ray at different ranges. However these estimates are undirect: for their interpretation we need to specify the A-Kdp relationship which depend on the precipitation type. In addition, the Phidp profiles are known to be noisy for low precipitation rates. Yes the Phidp used for the PIA estimates are the values near the mountain target (with a possible slight underestimation of the resulting polarimetric PIA compared to the MRT PIA which is determined over the entire range extent of the target). In our approach, we trust the MRT PIAs and, among other points, we use them for the interpretation of the Phidp measurements and the optimization of the A-Kdp coefficients.

In this study, the authors address the impact of attenuation in rain on the radial profiles of radar reflectivity at X band. Five different equations parameterized by the radar miscalibration error dC, radome attenuation PIA0, the error in the path-integrated attenuation PIAm estimated from the mountain radar signal, and the multipliers aAZ and aAK in the power-law $A - Z$ and $A - KDP$ relations are used to retrieve the unbiased radial profile of Z. Four of these equations are nonpolarimetric and three of them are constrained by PIAm whereas the fourth is a classic Hitschfeld-Bordan solution which is very unstable for higher values of PIAm. The four unknown parameters are varied in different combinations within certain ranges and the combination which yields the best match between 5 radial profiles of retrieved Z is considered a solution for all four parameters. The authors found that the estimated values of aAZ and aAK using their approach are consistent with the corresponding values derived from the simulations based on the DSD measurements and claimed this as a feasibility test of the method.

Thank you for the time / effort spent on reviewing this article and the good summary made above.

It is difficult to read this manuscript. There are too many parameters and equations.

We can understand this comment! We may consider presenting much of section 2 as appendices in the revised version (although this is not recommended in the AMT authors' guide). On the other side, we found important for the "young generation" to revisit in some detail the attenuation problem which is known to be severe at X-band and higher frequencies, and not negligible at lower frequencies (e.g at C-band). The mathematical formalism exposed is inspired by the seminal article of Marzoug and Amayenc (1994), two scientists I had the pleasure of interacting with some decades ago.

For example, I had hard time to realize that PIA0 and PIAm are identical to AF(r0) and AF(rm) expressed in a logarithmic scale.

This was explained in lines 135-141.

Equation (3.1) for the cost function is not understandable and requires more explanation. I guess that most of the readers (including myself) may not be familiar with the LHS technique and the Nash-Sutcliffe Efficiency (NSE) measure of the difference between two radial profiles of Z. These have to be defined and explained in a more detail as well as the terms OPS and NOPS.

OK, we will try to improve these points in revising section 3.1 which exposes actually the core of the methodology. Readers may be a little tired by the time they reach this point, all the more reason to try to lighten previous section 2 with appendices. The cost function is a simple mean of NSE coefficients calculated between pairs of reflectivity profiles corrected with the AZ algorithms (first 4 terms) and between pairs of PIA profiles involving the polarimetric algorithm (last two terms). The NSE is an interesting metrics (a kind of correlation coefficient) since it is sensitive to the correlation but also to the bias between the compared data. The LHS technique is quite common in computer experiments as a sampling method of multidimensional spaces of parameters. We used the lhs R package for its implementation. Finally, OPS stands for 'optimal parameter set", i.e. a set of parameters leading to a cost function value greater than a given satisfactory threshold, in other words a parameter set leading to a good convergence of the considered reflectivity and PIA profiles. NOPS stands for the "number of optimal parameter sets" obtained for a given simulation, e.g. for a particular event, for given values of the calibration error and the exponents of the A-Z and A-Kdp

relationships, for given ranges of variation of the LHS sampled parameters, etc. The NOPS is used for instance as a metrics to determine values of the calibration errors (Fig. 7) or to evidence the slight non-linearity of the A-Kdp relationship (Fig. 8).

It is not clear what is the ultimate purpose of the effort – more accurate QPE in the mountainous areas? Is there intention to estimate rain rate from corrected radial profiles of Z? It is well known that Z-based rainfall algorithms are not optimal and the methodologies based on KDP and A demonstrate much better performance, particularly at X band. It looks like using ZPHI-like retrievals of the radial profile of specific attenuation A and the R(A) relations is a more efficient and economic way to quantify rainfall. Moreover, the authors have benefit of determining the variable parameter α = A/KDP because they can directly measure the path-integrated attenuation PIAm using radar echoes form the mountains in their area along with a total span of differential phase ΔΦDP over the propagation path.

Thank you for these very interesting comments. Regarding the ultimate purpose (ambition!?) of the work, I would say that this article represents one step in the construction of an observational model dedicated to the estimation of atmospheric precipitation in all its forms (liquid, but also melting and solid) in a high mountain context with the rich observations collected within the RadAlp project. The idea is to formulate all available equations from all sources of information (backscattered power, polarimetry, … mountain returns!) in a rigorous mathematical framework and to consider the problem of parameter optimisation through a generalised sensitivity analysis (GSA) approach. By GSA, we mean considering the simultaneous effect of variations in all the parameters together and not the isolated effect of the variations of one particular parameter (e.g. the influence of the prefactor of the A-Kdp relationship on QPE). As proposed in this article for the "simple" case of convective precipitation with no ML contamination, this requires defining the structure of the parameters (inter-dependencies, a priori values, physical ranges of variation), exploring the parameter space with ad hoc sampling techniques, defining "cost functions" and "satisfaction thresholds" and performing the analysis of the statistical distributions of the a posteriori parameters. We have obtained encouraging results regarding the possibility to estimate calibration errors, radome attenuations, parameters of the A-Z and A-Kdp relationships (consistent with those derived independently from ground-based DSD measurements) using radar data alone. As noted at the end of the conclusion: "As a next step, we plan to extend the procedure to stratiform events with MOUC radar measurements made at times within or above the melting layer (added comment: i.e. in snow or melting precipitation; How can we estimate the coefficients of the A-Z-Kdp-R relationships for snow and melting precip?). The multi-angle, multi-frequency, polarimetric measurements of the valley-based radars will be critical in this respect for the characterization of the ML from below (Khanal et al. 2019, 2022) and the mitigation of the mathematical ambiguity of the physical model of interest". We may add that such a procedure is likely to be difficult to implement in an operational context due to its principles and computational costs. It may find its utility for the parameterization of radar QPE algorithms as well as for post-event analyses.

This manuscript describes a combined QPE algorithm based on polarimetric X-band radar measurements as applied in the mountainous terrain. I suggest that the authors revise the manuscript having in mind comments below.

Thank you for the time spent on this review and for the valuable comments made

It would be beneficial for this paper if you can provide some quantitative information on how the results of your algorithm developments and attenuation estimates would benefit the accuracy of QPE retrievals compared to existing algorithms.

As explained in our reply to reviewer 2's last comment, "this article represents one step in the construction of an observational model dedicated to the estimation of atmospheric precipitation in all its forms (liquid, but also melting and solid) in a high mountain context with the rich observations collected within the RadAlp project. The idea is to formulate all available equations from all sources of information (backscattered power, polarimetry, … mountain returns!) in a rigorous mathematical framework and to consider the problem of parameter optimisation through a generalised sensitivity analysis (GSA) approach. By GSA, we mean considering the simultaneous effect of variations in all the parameters together and not the isolated effect of the variations of one particular parameter".

In other words, rather than a "combined QPE algorithm", we are proposing a procedure for optimising the parameters of the equations describing the attenuation physics, the results of which could be used in an operational QPE system.

Since the article is already rather complex and lengthy, we have made the choice in the first version to focus on the estimation of the coefficients A-Z and A-Kdp relationships, the calibration error and the radome attenuation by using the MRT PIA measurements, and to leave aside the following problem of rainrate estimation.

Within the final stages of his PhD, Anil Kumar Khanal is being performing an assessment of various QPE algorithms using the parameterizations obtained in this article with respect to independent raingauge data. Including such forthcoming results in the revised version of the article may be an option we have to consider.

Would a simple R-Kdp based QPE method still have important advantages? This method is insensitive to attenuation and to the radar absolute calibration errors and may be preferential for moderate and heavier rainfall.

The R-Kdp method is one of the good candidates for QPE. In our context, its parameterization for liquid precipitation could be based on the available DSD measurements. The ZPHI method (Testud et al. 2000) has also very interesting properties (independence on dC, akz, on-site attenuation). Being a "rain-profiling algorithm" using both reflectivity and phidp profiles, we have a preference for the latter polarimetric method due

to the noisiness of the phidp measurements for low to moderate rainfall. However, one more time, comparing QPE algorithms is not our goal in this article: we are using attenuated reflectivity profiles, Phidp profiles and mountain-derived PIA estimates to optimise (some of) the parameters of the attenuation equations.

Optimizing the R-Kdp and R-A parameters would require including the additional raingauge measurements in the GSA framework. This is certainly desirable but not implemented in the current version.

Mountain references are available only for a fraction of the radar beams. Please be more specific about how these limitations influence you approach.

We are not "promoting" MRT-constrained A-Z algorithms over polarimetric algorithms. We just want to outline that there is some valuable information in the mountain returns. Similarly, in satellite configurations, we could claim that the Surface Reference Technique brings additional information wrt dual-frequency measurements of the GPM core platform.

However, in some vulnerable valleys in high-mountain regions, one could well get "belts" of mountain targets allowing implementation of MRT-constrained A-Z algorithms in a continuous manner in the inner domain.

For convenience, our down-valley based X-band radar (XPORT) has been set up on the roof of the laboratory, but we have in mind a number of locations where it could be installed for an effective precipitation monitoring over the entire city of Grenoble with the MRT approach.

Does mountain reflectivity depend on the wetness of the ground targets? If yes, then "dry environment" reference measurements are not exactly applicable to rainy conditions.

This problem has been addressed in several previous publications of our team, the most informative being probably https://doi.org/10.1175/1520-0426(1999)016<0405:RMIHTW>2.0.CO;2, 1999 and more recently https://doi.org/10.5194/amt-13-3731-2020, 2020.

As explained in the article, (i) selecting strong mountain returns (typically greater than 45-50 dBZ) allows to mitigate the impact of precipitation falling over the target (negative bias), (ii) a refined estimation of the so-called dry-weather baseline is required to account for the possible modification of backscattering properties of the mountain surfaces before and after the event and (iii) the time variability of the dry-weather returns defines the minimum detectable PIA.

What is natural variability of the coefficients in A-Z and A-Kdp relations (eg, due the rain type – convective vs stratiform)?

Some information is available in the article on this subject from the DSD-derived relationships presented in Figs 5 and 6 with comments made in the text (lines 460-485). We actually used the DSD-derived scatterplots between the variables of interest for all the types

of precipitation observed in the Grenoble area to define the central values and the ranges of variation of the exponents and prefactors in the GSA procedure.

We have to add that in the assessment of the QPE algorithms wrt raingauge data that is being performed, we found necessary to consider an R-A relationship fitted on convective precipitation rather the one fitted over all precipitation types, due to a missfit of the highest rainrates in the latter.

There has been a significant number of studies deriving X-band A-Kdp relations using different approaches including model calculations and also the direct use of observational data (e.g., Bringi and Chandrasekar 2001 book, https://doi.org/10.1175/JTECH1763.1 https://doi.org/10.1175/JTECH1804.1 https://doi.org/10.1175/JTECH-D-13-00231.1 to name a few). It would be appropriate to compare (at least briefly) your relations with previous ones and also to provide a measure of uncertainty in the coefficients of these relations.

Yes! We recognize that such references and comments are missing. We will include them in the revised version.

Line 591: I believe it is "backscatter phase shift" not "phase shift on propagation". Also, non-uniform beam filling affects other approaches not only a polarimetric one.

Yes for the first point, thanks. Yes also for the second one!

Fig.2: There is a wedge of the high Phidp increase indicating heavier rainfall (17:00 UTC). However, (unlike for the Phidp wedge at 16:05 UTC) there is no corresponding high reflectivity areas (even in the closest to the radar range gates within the high intensity cell, where total attenuation is expected to be not yet significant). Please explain. Also, adding SNR frames can help to better interpret Fig. 2 data.

This is due to 2 factors: first, the big radome attenuation evaluated to about 10-15 dB and secondly the displayed reflectivity range which is limited to 10-60 dBZ. A complementary display can be found with the 2 examples of profiles in Fig. 9 within and outside the wedge. We will extend the display reflectivity range to [-10, 60 dBZ]

It appears that there are pixels (and clusters of pixels) of high rhohv values (at 14:00 UTC), which are not associated neither with rain cells in the Zm graph nor with the mountain slope echos. Please explain.

Light rain? Again, this may be related to the effect of the 10 dBZ lower limit for the displayed reflectivity.

Figs. 1 and 2: Please, increase the font size of numbers in the graph axes and color bars (currently the numbers are impossible to read) and show units on the color bar (e.g., dBZ in Fig. 1, right frame).

This will be done.